# Can LLM Already Serve as A Database Interface? A BIg Bench for Large-Scale Database Grounded Text-to-SQLs

**Jinyang Li**[1,♣,‡] **Binyuan Hui**[2,♣], **Ge Qu**[1,♣], **Jiaxi Yang**[2], **Binhua Li**[2], **Bowen Li**[6],
**Bailin Wang**[5], **Bowen Qin**[2], **Ruiying Geng**[2], **Nan Huo**[1], **Xuanhe Zhou**[3], **Chenhao Ma**[6],
**Guoliang Li**[3], **Kevin C.C. Chang**[7], **Fei Huang**[2], **Reynold Cheng**[1], **Yongbin Li**[2]

[1] The University of Hong Kong [2] DAMO Academy, Alibaba Group
[3] Tsinghua University [4] Shanghai AI Laboratory [5] MIT CSAIL
[6] The Chinese University of Hong Kong, Shenzhen
[7] University of Illinois at Urbana-Champaign
{jl0725,quge}@connect.hku.hk, ckcheng@cs.hku.hk
binyuan.hby@alibaba-inc.com

## Abstract

Text-to-SQL parsing, which aims at converting natural language questions into executable SQLs, has gained increasing attention in recent years. In particular, GPT-4 and Claude-2 have shown impressive results in this task. However, most of the prevalent benchmarks, i.e., Spider, and WikiSQL, focus on database schema with few rows of database values leaving the gap between academic study and real-world applications. To mitigate this gap, we present **BIRD**, a **BI**g bench for la**R**ge-scale **D**atabase grounded in text-to-SQL tasks, containing **12,751** text-to-SQL pairs and **95** databases with a total size of **33.4 GB**, spanning **37** professional domains. Our emphasis on database values highlights the new challenges of dirty and noisy database values, external knowledge grounding between NL questions and database values, and SQL efficiency, particularly in the context of massive databases. To solve these problems, text-to-SQL models must feature database value comprehension in addition to semantic parsing. The experimental results demonstrate the significance of database values in generating accurate text-to-SQLs for big databases. Furthermore, even the most effective text-to-SQL models, i.e. GPT-4, only achieve 54.89% in execution accuracy, which is still far from the human result of 92.96%, proving that challenges still stand. We also provide an efficiency analysis to offer insights into generating text-to-efficient-SQLs that are beneficial to industries. We believe that BIRD will contribute to advancing real-world applications of text-to-SQL research. The leaderboard and source code are available: https://bird-bench.github.io/.

## 1 Introduction

Text-to-SQL parsing [55, 50, 51, 3, 52, 37], which focuses on converting natural language questions into SQL queries, has attracted significant research interests from both academia and industry. This attention stems from its potential to empower non-expert data analysts in automatically extracting desired information from ubiquitous relational databases using natural language. Recent advances in neural models, including those based on large language models (LLMs), have led to an impressive performance on existing benchmarks such as SPIDER [53] and WikiSQL [58]. For instance, the execution accuracy of the top-performing model in the SPIDER leaderboard has increased from 53.5%

---

♣ Equal contribution.
‡ Work done during an intern at Alibaba DAMO Academy.

37th Conference on Neural Information Processing Systems (NeurIPS 2023) Track on Datasets and Benchmarks.

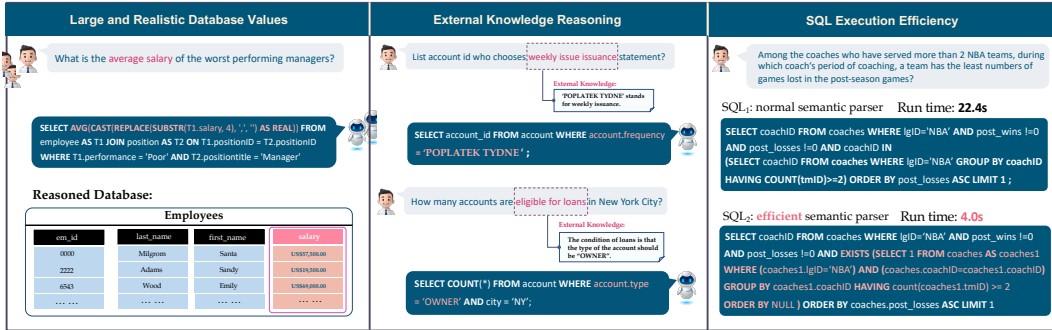

Figure 1: Examples of challenges in our **BIRD** benchmark. 1) databases contain values of noisy data types [14, 23, 19, 31]. In the left example, the average salary could be fetched by processing the data type from string (`TEXT` in SQLite) to float (`REAL` in SQLite) after deleting the special tokens, `"US$"` and `","`. 2) external knowledge and reasoning are required. In the middle example, models must handle that only `"OWNER"` accounts are eligible for loans. 3) query execution efficiency needs to be considered. In the right example, the adoption of more efficient SQL queries leads to significant gains in speed, which is of great value in industries.

[59] to 85.3% [35] over the past three years. The latest SOTA parser [35] in SPIDER benefits from the powerful understanding and coding capabilities of the large language model (LLM), and such excellent performance leads us to ask a question: ***Can LLM already serve as a database interface ?***

The answer is no, as previous benchmarks focus on database schema with few rows of database values leaving the gap between academic study and the real world. As shown in Figure 1, first, we discovered that current state-of-the-art models still struggle to generalize to more realistic situations characterized by large database sizes and noisy values. Second, the growth in database sizes often results in much context compression, making it challenging to reveal the entire context [1]. Thus it requires external knowledge reasoning for a comprehensive understanding. Third, existing benchmarks do not account for SQL execution efficiency, which holds significant practical importance in real-life applications, notably in the case of large databases. Motivated by these observations, we aim to develop a new text-to-SQL benchmark that better represents real-life scenarios and narrows the gap between experimental and practical settings.

In this work, we propose **BIRD**, a **BI**g Bench for La**R**ge-Scale **D**atabase Grounded in Text-to-SQLs for real-world applications. BIRD contains complex **12,751** examples of querying information over **95** big databases with a total size of **33.4 GB** spanning **37** professional domains. For training and development, we collected and modified 80 open-source relational databases from real analysis platforms (Kaggle, Relation.vit). To further avoid data leakage, we curated 15 additional relational databases for a hidden test set. Given these databases, we rely on crowdsourcing to collect natural language questions and the corresponding SQLs. Additionally, we propose a new evaluation metric Valid Efficiency Score (VES) to evaluate the efficiency of generated SQLs. To the best of our knowledge, BIRD is the first text-to-SQL benchmark to incorporate efficiency, promoting more efficient query methods within the context of massive and noisy database values.

We evaluate the performance of state-of-the-art text-to-SQL parsers using two popular methodologies: fine-tuning (FT) with T5 [38], and in-context learning (ICL) with advanced large language models (LLMs) such as ChatGPT [33] (`gpt-3.5-turbo`), Claude-2 [2] (`claude-2.0`), GPT-4 [32] (`gpt-4-32k`). Our experimental results demonstrate that the current models struggle to generalize well on BIRD. Specifically, even the GPT-4 only achieves 54.89% in execution accuracy. In comparison, the performance still lags far behind the human performance of 92.96%, proving that challenges still stand. Moreover, we perform a comprehensive analysis to provide insight and direction. We encourage further research by the NLP and DB communities to jointly address the more realistic settings presented in this benchmark.

## 2 Task Formulation & Annotations

Text-to-SQL refers to the process of converting a natural language question $\mathcal{Q}$ into a SQL query $\mathbf{Y}$ capable of retrieving relevant data from a database. The database can be represented as $\mathcal{D} = \langle \mathcal{C}, \mathcal{T} \rangle$,

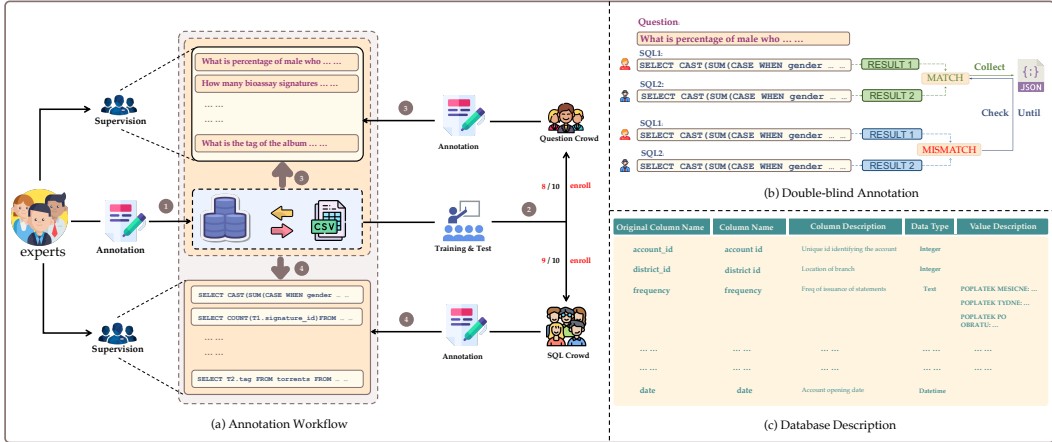

Figure 2: An Overview of the BIRD Annotation Workflow in (a). This figure depicts a four-step procedure. (1) The workflow begins with specialists assembling and producing databases and description files. (2) Experts then teach and evaluate crowdsourcing people, keeping only those who pass the evaluation. (3) Question annotators create a corpus of questions using databases and their corresponding description files. (4) SQL annotators produce SQL files, equipped with databases, descriptions, and questions. (b) and (c) also depict the Double-blind annotation procedure and an example of database descriptions.

where $\mathcal{C}$ and $\mathcal{T}$ are columns and tables respectively. When dealing with complex database values, such as BIRD, it is crucial to incorporate external knowledge evidence, denoted as $\mathcal{K}$, to improve the models' understanding of database values. Finally, the text-to-SQL could be formulated as:

$$\mathbf{Y} = f(\mathcal{Q}, \mathcal{D}, \mathcal{K} \mid \boldsymbol{\theta}), \tag{1}$$

where the function $f(\cdot \mid \boldsymbol{\theta})$ can represent a model or neural network with the parameter $\boldsymbol{\theta}$.

## 3 Dataset Construction

### 3.1 Annotation Entrance

To deliver a high-quality benchmark, we administer thorough exams to all applicants and only hire those who pass these rigorous tests. Further information is available in the Appendix B.2.

### 3.2 Database Source

It is difficult to collect databases with complex schemas and sufficient value due to privacy protection. Earlier works [45, 53] choose to self-design database schemas and value production. Nonetheless, the value distribution and schemas may differ from real-world scenarios in this way. In our work, we obtain and process databases from three different sources to enrich real-world attributes. 32% of our databases are sourced from Kaggle*, a platform renowned for holding data science competitions with difficult, noisy values and schemas. Another 48% come from CTU Prague Relational Learning Repository†, an open platform for machine learning research with multi-relational data. The remaining 20% are built by acquiring open tables, synthesizing and standardizing schemas, and generating database constraints. All of these databases contain real and large value distributions and are easily accessible with the appropriate licenses. Finally, we present 95 databases consisting of 69, 11, and 15 databases for training, development, and testing respectively. Our databases cover 37 professional domains, including blockchain, sports, health care, politics, etc. We anticipate that it will be a significant resource for researchers to explore domain generalization in semantic parsing tasks with large database values.

---

*https://www.kaggle.com/
†https://relational.fit.cvut.cz/

### 3.3 Question Annotation

**Database Description File.** The Database Description File is a crucial resource designed to aid annotators in comprehending database values, thereby allowing them to ask insightful questions. It offers two primary pieces of information regarding the database. **(1) Full schema names:** database table and column names are frequently abbreviated, which are difficult to understand. **(2) Value description:** this aspect is particularly useful when phrases or tokens in a question do not directly match values in the database.

**External Knowledge Evidence.** In our study of professional data analysis, we find that external knowledge evidence is required to map the natural language instructions into counterpart database values. Therefore, we collect and classify such evidence into four categories: **(1) Numeric Reasoning Knowledge:** this category refers to the mathematical computation required for certain SQL operations. In our benchmark, we present 8 basic math operations, including 4 complex operations as [7]: MINUS, ADDITION, DIVISION, MULTIPLY. BIRD also contains compositional operations over basic ones, such as percentages, formulas, etc. **(2) Domain Knowledge:** this category consists of domain-specific knowledge that is utilized to generate SQL operations [10, 57]. For instance, a business analyst in the banking business requires knowledge of financial indicators such as return on investment and net income in order to generate effective SQL queries. **(3) Synonym Knowledge:** this category includes words or expressions that have the same or similar meanings regardless of how they are phrased differently [11]. **(4) Value Illustration:** this category refers to detailed descriptions of database values, including value types, value categories, and the mapping combinations of columns and values that correspond to entities, for example: "center" can be represented by "pos = C" in the database professional_basketball.

### 3.4 SQL Annotation

**Double-Blind Annotation.** As shown in Figure 2 (b), we employ a double-blind approach [42] for SQL annotation. This approach involves two independent SQL annotators who generate SQLs for the same question without discussion. The annotated SQLs are executed in databases, and those yielding identical results are gathered. Otherwise, the SQLs are checked with experts until a consensus is reached. Double-blind procedures can dramatically reduce the SQL annotation error rate, as there is a small probability for two skillful annotators to generate the same incorrect results when databases have large values. The more semantic-equivalent and efficient SQL selected by experts for each question is picked as ground truth SQL in BIRD, and the external knowledge evidence sentences are recorded for each SQL if utilized.

**Examination.** Experts evaluate each text-to-SQL pair to ensure the highest quality of data. The evaluation process includes two dimensions: SQL validness, and text-knowledge-SQL alignment. Firstly, the SQL validness will be confirmed that each SQL is executable and can return a valid result from the database. The "valid result" refers to the set of results that is not "NULL". If the executed result set is "NULL", experts will make slight changes to the conditions of the questions until the associated SQLs can provide a valid result set. Secondly, text-knowledge-SQL alignment is involved to ensure that each SQL can be generated with the given texts and knowledge evidence. If the evidence is insufficient to generate the SQL or contains errors, experts will be in charge of correcting them.

## 4 Data Statistics

**Overall Statistics** Table 1 presents an overview comparison between BIRD and other cross-domain text-to-SQL benchmarks. As the statistics demonstrate, BIRD is a large-scale cross-domain benchmark, covering complex SQL functions, knowledge reasoning, and efficiency evaluation.

**Question Statistics** Database values bring more challenges in text-to-SQLs. In order to underscore this, we classify questions into two macro-categories: Fundamental Type and Reasoning Type, and each contains 4-5 micro-categories in detail. The Fundamental Type of questions refers to those that can be answered without database value comprehension. It contains Match-based (83.9%), Ranking (20.3%), Comparison (16.7%), Counting (30.4%), Aggregation (15.7%). The

Table 1: An overview comparison between BIRD and other cross-domain text-to-SQL benchmarks. In SQL, `Function` pertains to the SQL functions (Appendix B.11). `Knowledge` refers to whether or not this dataset necessitates external knowledge reasoning from the model. `Efficiency` refers to whether or not this dataset takes into consideration execution efficiency.

| Dataset | # Example | # DB | # Table/DB | # Row/DB | Function | Knowledge | Efficiency |
|---|---|---|---|---|---|---|---|
| WikiSQL [58] | 80,654 | 26,521 | 1 | 17 | ✗ | ✗ | ✗ |
| SPIDER [53] | 10,181 | 200 | 5.1 | 2K | ✗ | ✗ | ✗ |
| KaggleDBQA [24] | 272 | 8 | 2.3 | 280K | ✗ | ✓ | ✗ |
| BIRD | 12,751 | 95 | 7.3 | 549K | ✓ | ✓ | ✓ |

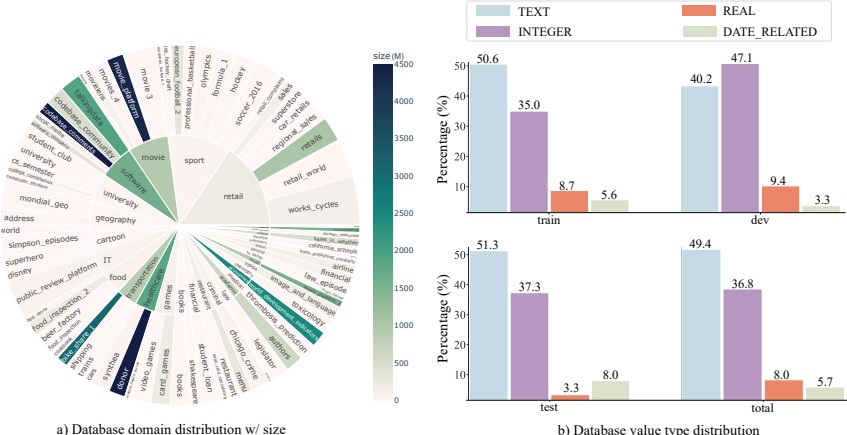

a) Database domain distribution w/ size  b) Database value type distribution

Figure 3: This is a comprehensive database distribution in the BIRD. a) shows the domain and size distribution of each database. And b) shows the data type distribution of databases.

`Reasoning Type` entails questions that demand the external knowledge grounding on values, which is exclusive to BIRD. To be specific, the questions about `Domain Knowledge` (23.6%), `Numeric Computing` (24.5%), `Synonym` (7.2%), `Value Illustration` (70.1%), are involved in BIRD. There are ample examples in Appendix B.3. In addition, we observe that 70.1% of the questions need value illustrations. This indicates that more real-world questions in text-to-SQL applications demand a thorough understanding of database values, which is consistent with our motivation for creating the BIRD benchmark.

**Database Statistics**   In BIRD, we investigate the distribution of database domains, database size, and value types. Figure 3 (a) presents a detailed distribution of domains and their counterpart databases in a sunburst diagram for both training and development sets. The area of each semi-circle corresponds to the number of text-to-SQL pairs in this database. Figure 3 (a) also shows the size distributions of databases. The darker color means a larger size of databases, and vice versa. For example, the database `Donor` is the largest database with 4.5 GB in this dataset. Furthermore, we observe from Figure 3 (b) that a considerable proportion of BIRD's data comprises date-related values. Considering that real-world applications often rely on time-sensitive data [25], the prevalence of such questions highlights the practical purposes.

**SQL Statistics**   We provide the complexity and diversity of SQLs in BIRD. As illustrated in Figure 4, we present a comprehensive distribution analysis of SQLs across four dimensions. `No.Toks / SQL` and `No.JOINs / SQL` demonstrate the intricacy of the SQLs in BIRD. `No.of Keywords` and `No.n-grams / SQL (n=3)` serve as the support for the diverse patterns of SQLs since we decouple the question and SQL annotation procedures to make the situation more realistic [6].

## 5   Evaluation Metrics

In contexts of practical data analysis, text-to-SQL models are prioritized for delivering expected results accurately and efficiently. Thus we provide two metrics in BIRD, execution accuracy (EX)

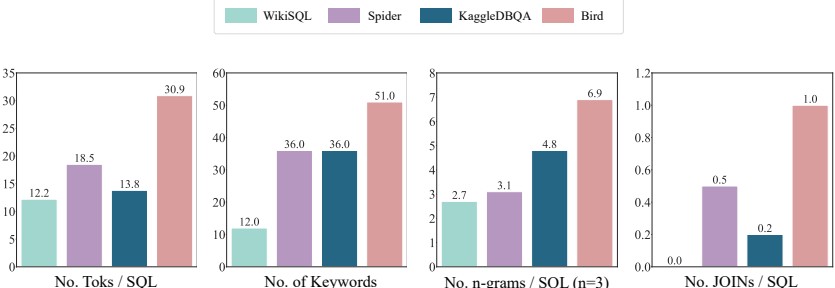

Figure 4: A comparative statistical analysis of SQL queries in the BIRD dataset and other cross-domain text-to-SQL benchmarks.

and valid efficiency score (VES) to evaluate text-to-SQL parsers confronted with large real-world database values.

**Execution Accuracy (EX)**   EX is defined as the proportion of examples in the evaluation set for which the executed results of both the predicted and ground-truth SQLs are identical, relative to the overall number of SQLs [37]. Considering the result set as $V_n$ executed by the $n^{th}$ ground-truth SQL $Y_n$, and the result set $\hat{V}_n$ executed by the predicted SQL $\hat{Y}_n$, EX can be computed by:

$$\text{EX} = \frac{\Sigma_{n=1}^{N} \mathbb{1}(V_n, \hat{V}_n)}{N}, \tag{2}$$

where $\mathbb{1}(\cdot)$ is an indicator function, which can be represented as:

$$\mathbb{1}(V, \hat{V}) = \begin{cases} 1, & V = \hat{V} \\ 0, & V \neq \hat{V} \end{cases} \tag{3}$$

**Valid Efficiency Score (VES)**   VES is designed to measure the efficiency of valid SQLs generated by models. It is worth noting that the term "valid SQLs" refers to predicted SQL queries whose result sets align with those of the ground-truth SQLs. Any SQL queries that fail to fetch the correct values will be declared invalid since they are totally useless if they cannot fulfill the user requests, regardless of their efficiency. In this case, the VES metric considers both the efficiency and accuracy of execution results, providing a comprehensive evaluation of a model's performance. Formally, the VES can be expressed as:

$$\text{VES} = \frac{\Sigma_{n=1}^{N} \mathbb{1}(V_n, \hat{V}_n) \cdot \mathbf{R}(Y_n, \hat{Y}_n)}{N}, \quad \mathbf{R}(Y_n, \hat{Y}_n) = \sqrt{\frac{\mathbf{E}(Y_n)}{\mathbf{E}(\hat{Y}_n)}} \tag{4}$$

where $\mathbf{R}(\cdot)$ denotes the relative execution efficiency of predicted SQL in comparison to ground-truth SQL, allowing for machine status-related uncertainty. $\mathbf{E}(\cdot)$ is a function to measure the absolute execution efficiency for each SQL in a given environment[‡]. Furthermore, we incorporate the square root function to minimize random instances that are abnormally faster or slower than the ground-truth SQLs. Here, efficiency can refer to running time, throughput, memory cost, or merged metrics. In BIRD, we consider the running time mainly at this time. Appendix B.8 provides a detailed description of the VES.

## 6 Experiments

### 6.1 Baseline Models

We present the performance of two types of baseline models in BIRD. The first type of model is based on fine-tuning (FT) techniques, which outputs SQL by tuning all parameters of language models to learn the annotated train set. On the other hand, the second type of model based on

---

[‡]In BIRD evaluation, we run 100 times for each SQL in the same CPU and evaluate average results after dropping the outliers.

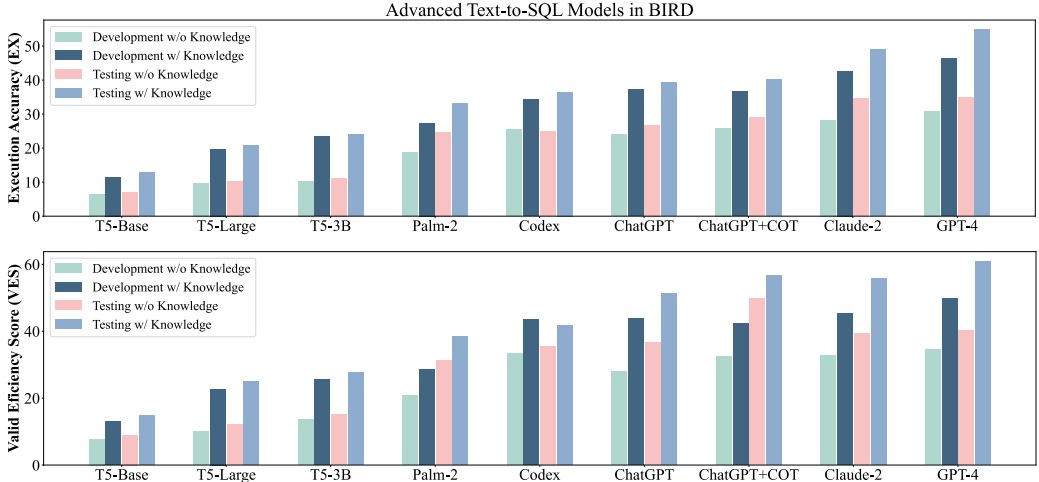

Figure 5: A bar chart provides a clear visualization of the performance of advanced models on BIRD.

Table 2: The Execution Accuracy (EX) of advanced text-to-SQL models in BIRD. The human performance is also provided.

| Models | Development Data | | Testing Data | |
|---|---|---|---|---|
| | w/o knowledge | w/ knowledge | w/o knowledge | w/ knowledge |
| | *FT-based* | | | |
| T5-Base | 6.32 | 11.54 (+5.22) | 7.06 | 12.89 (+5.83) |
| T5-Large | 9.71 | 19.75 (+10.04) | 10.38 | 20.94 (+10.56) |
| T5-3B | 10.37 | 23.34 (+12.97) | 11.17 | 24.05 (+12.88) |
| | *ICL-based* | | | |
| Palm-2 | 18.77 | 27.38 (+8.61) | 24.71 | 33.04 (+8.33) |
| Codex | 25.42 | 34.35 (+8.93) | 24.86 | 36.47 (+11.61) |
| ChatGPT | 24.05 | 37.22 (+13.17) | 26.77 | 39.30 (+12.53) |
| ChatGPT + COT | 25.88 | 36.64 (+10.76) | 28.95 | 40.08 (+11.24) |
| Claude-2 | 28.29 | 42.70 (+14.41) | 34.60 | 49.02 (+14.42) |
| GPT-4 | **30.90** | 46.35 (+15.45) | 34.88 | 54.89 (+20.01) |
| GPT-4 + DIN-SQL | - | **50.72** | - | 55.90 |
| Human Performance | - | - | **72.37** | **92.96** (+20.59) |

in-context learning (ICL), can generate results without additional training. In FT models, we select T5 family [38] as the main baseline models. For ICL-based models, we provide zero-shot results of Codex (`code-davinci-002`), ChatGPT (`gpt-3.5-turbo`), GPT-4 (`gpt-4-32k`), Claude-2 (`claude-2.0`), Palm-2 (`text-bison-001`). Additionally, we also implement a state-of-the-art (SOTA) model of SPIDER, DIN-SQL [35], to evaluate the challenges proposed by the BIRD dataset. Table 2, Table 3 and Figure 5 present the overall results of advanced language models on BIRD.

## 6.2 Execution Accuracy Analysis

Table 2 and Figure 5 presents stratified performances of various models in BIRD. GPT-4 surpasses all baseline language models. Claude-2 closely follows, demonstrating outstanding abilities in semantic parsing and knowledge reasoning. Further, the incorporation of a dedicated reasoning prompt by [35], enables DIN-SQL + GPT-4 to achieve a new state-of-the-art result on BIRD. It contains value sampling, few-shot demonstrations, and self-correction. Despite considerable advancements in Language Model Learning (LLMs) and prompt intelligence, the performance of these models lags obviously behind human capabilities. Not only does this gap highlight the complex nature of BIRD, but it also presents opportunities for uncovering more capable models or advanced reasoning prompt methods applicable to real-world text-to-SQL scenarios.

Table 3: The Valid Efficiency Score (VES) of advanced text-to-SQL models in BIRD. The human performance is also presented.

| Models | Development Data | | Testing Data | |
|---|---|---|---|---|
| | w/o knowledge | w/ knowledge | w/o knowledge | w/ knowledge |
| *FT-based* | | | | |
| T5-Base | 7.78 | 12.90 (+5.12) | 8.97 | 14.71 (+5.74) |
| T5-Large | 9.90 | 22.74 (+12.84) | 12.25 | 25.00 (+12.75) |
| T5-3B | 13.62 | 25.57 (+11.95) | 15.17 | 27.80 (+12.63) |
| *ICL-based* | | | | |
| Palm-2 | 20.82 | 28.64 (+7.82) | 31.32 | 38.41 (+7.09) |
| Codex | 33.37 | 43.41 (+10.04) | 35.40 | 41.60 (+6.20) |
| ChatGPT | 27.97 | 43.81 (+15.84) | 36.68 | 51.40 (+14.72) |
| ChatGPT + COT | 32.33 | 42.30 (+9.97) | 49.69 | 56.56 (+6.87) |
| Claude-2 | 32.75 | 45.28 (+12.53) | 39.32 | 55.77 (+16.45) |
| GPT-4 | **34.60** | 49.77 (+15.17) | 40.20 | 60.77 (+20.57) |
| GPT-4 + DIN-SQL | - | **58.79** | - | 59.44 |
| Human Performance | - | - | **70.36** | **90.27** (+19.91) |

## 6.3 Baseline Performance on Spider

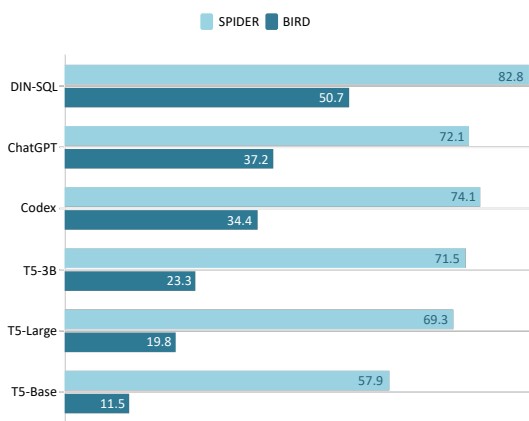

Figure 6: The EX results of the same baseline models on the SPIDER and BIRD dev set.

SPIDER [53] is the most prevalent and complex cross-domain text-to-SQL benchmark. It mainly focuses on evaluating schema-relevant semantic parsing capabilities. To demonstrate the increasing difficulty of the BIRD dataset due to its complex database schema and values, we visualize the execution accuracy of the same baseline models on both the BIRD and SPIDER datasets. To ensure a fair evaluation, all models are furnished with knowledge about values, and the same programming prompt is implemented for Language Models (LMs) across the two datasets. Figure 6 shows the concentration on database values makes BIRD become the most challenging text-to-SQL benchmark. This disparity in the performance of each model demonstrates the need for further research and development of models capable of handling complicated database schema and values.

## 6.4 Efficiency Analysis

According to Table 3, we can observe that models with higher EX can more possibly achieve higher VES. This can be explained by the prerequisite that text-to-SQL models must accurately predict results in order to attain a higher VES, which fulfills the practical purpose.

**Two-Stage Optimization.** Intuitively, the goal of text-to-efficient-SQL conversion can be decomposed into two sub-stages. Following previous text-to-SQL tasks, the first sub-stage, semantic parsing, concentrates on accurately converting questions into SQL queries. The second sub-stage involves optimizing the SQL queries, rewriting them to be more efficient while maintaining the same results [61]. To demonstrate the efficacy of this approach, we selected 10 random examples from the development set where ChatGPT accurately predicted the results. Then, our specialists optimize these queries based on the established query optimization rules [28, 34, 62]. We observe that the two-stage optimization leads to an average time-saving of 77.75% while keeping the same results.

**Chat w/ Database.** BIRD introduces the novel mode of "Chat With Database", which enables models to be aware of data types and distributions by generating global SQL queries that interact with databases. This approach lays the foundation for the development of more effective and efficient SQL queries. As observed in the experiment, the time-saving percentage of the SQL queries can

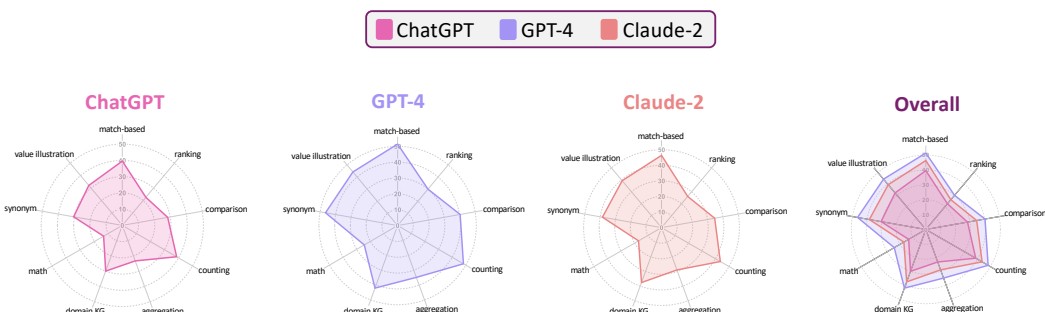

Figure 7: The fine-grained categorical evaluation of advanced large language models on BIRD.

be reached at 87.3% by configuring indexes within the database. The detailed efficiency analysis is presented in Appendix B.5.

## 6.5 Knowledge Evidence Analysis

We implement each baseline model for both two scenarios. The first is **NOT** to provide the ground truth external knowledge evidence sentence (w/o knowledge) for each sample. The other testing bed is to provide such evidence (w/ knowledge) and make text-to-SQL models do knowledge grounding by themselves. As we discuss in Section 3.3, expert annotations on external knowledge evidence sentences are employed to enhance the model's comprehension of database values.

After being easily fed with the external knowledge evidence about the database values, all models have a clear improvement across the different difficulty levels as shown in Table 2 and Table 4. ***This indicates that external knowledge evidence in BIRD is effective and instructive for models to better understand the database values.*** Also it illustrates that the database values are very important to text-to-SQL models when facing more real databases. Besides, ICL-based approaches have a better self-knowledge grounding capability and pre-trained SQL knowledge than FT smaller models with less than 5B parameters. Equipped with COT, ChatGPT can perform better, since multi-step reasoning is beneficial when the knowledge and data are low-resource. Despite this, we observe a decline or limited improvements in performance for ChatGPT + external knowledge evidence for COT version. We hypothesize that the internal multi-step knowledge reasoning of LLMs is not compatible with the way of external knowledge (evidence) in this situation. Therefore, the development of methods that effectively combine the strong multi-step self-reasoning capabilities of LLMs with external knowledge reasoning coherently presents a promising future direction [29].

## 6.6 More Analysis

**Fine-grained Category Analysis.** Figure 7 provides a detailed comparison of various dimensions of sub-capabilities of advanced LLMs on BIRD. The results indicate that GPT-4 exhibits superior performance against ChatGPT and Claude-2 in all areas. Nevertheless, there is a notable disparity in the performance of ranking and numerical computing (math) among all the models. This limitation may suggest the inadequacy of contemporary LLMs for deep data science tasks because such tasks always incorporate mathematical computations and rankings within the context of vague user queries. Conversely, these models demonstrate relatively better performance in domain knowledge, synonym detection, and value illustration, which can be attributed to their adequate linguistic training and reasoning capabilities during the pretraining phases.

**Human Performance.** In order to activate the efforts of text-to-SQL studies to achieve an application-level performance in real-world scenarios, we provide human performance in BIRD. Table 2, Table 3 shows that there's still a huge gap between even SOTA text-to-SQL models and human performance. The thorough introduction of procedures is in Appendix B.9.

**Error Analysis.** ChatGPT is currently the most prevalent and cost-efficient LLM. Therefore, the performance of ChatGPT is concentrated in this error analysis. The detailed analysis is in Appendix B.6. We observe 500 randomly sampled error cases, providing an in-depth assessment in the following

categories. **Wrong Schema Linking (41.6%)** pertains to the scenario where ChatGPT can accurately comprehend the structure of the database but erroneously associates it with inappropriate columns and tables. This demonstrates that the task of schema linking [43, 57], even in intricate and practical situations, continues to be a significant obstacle for models. **Misunderstanding Database Content (40.8%)** occurs when ChatGPT either fails to recall the correct database structure (e.g., `rtype` doesn't belong to the `satscores` table) or generates fake schema items (e.g., `lap_records` is not appearing in the `formula_1` database and many values are predicted incorrectly) especially when the database is very large. In this case, how to make ChatGPT really understand database structure and values [27] is still a pain point topic in LLMs. **Misunderstanding Knowledge Evidence (17.6%)** refers to cases in which the model does not accurately interpret human-annotated evidence. An instance is that ChatGPT directly copies the formula `DIVIDE(SUM(spent), COUNT(spent))`. This finding demonstrates that ChatGPT exhibits a lack of robustness in response to unfamiliar prompts or knowledge, causing it to directly replicate formulas without considering SQL syntax [15]. We also observe that ChatGPT occasionally employs incorrect keywords (e.g., misusing the MySQL `Year()` function instead of an SQLite function `STRFTIME()`) or exhibits decoding errors.

## 7  Related Work

High-quality datasets are crucial for advancing various natural language processing tasks, including text-to-SQL. Early single-domain text-to-SQL datasets like GeoQuery [55], ATIS [9], and Restaurant [20] targeted specific information retrieval tasks, while more recent datasets such as WikiSQL [58] and SPIDER [53] propose cross-domain dataset to require domain generalization. However, most cross-domain text-to-SQL datasets still emphasize database schema rather than values, diverging from real-world scenarios. KaggleDBQA [24] addressed this by constructing 272 text-to-SQL pairs from eight databases on Kaggle, while other datasets like EHRSQL [25], SEDE [13], and MIMICSQL [46] collected diverse, large-value databases with more professional SQL queries. Despite these advancements, these datasets remain single-domain focused. Recent work has explored knowledge-intensive text-to-SQL benchmarks [10, 57], aiding experts in real-world analysis through knowledge grounding. BIRD is the first large-scale benchmark to incorporate these real-world features, emphasizing database values.

## 8  Limitation and Future work

Despite the high quality of SQL annotation produced by double-blind annotation, the procedure is resource-intensive. Future research could explore a human-computer interaction (HCI) based approach, incorporating advanced AI systems such as GPT-4 for taking parts of annotation duties, to maintain data quality while reducing human effort. In addition, SQLite was chosen as the primary SQL codebase for previous text-to-SQL benchmarks and this study since it's friendly to users. While it presents difficulties in fetching Query Execution Plans (QEP) for precise efficiency computation and adapting to different SQL syntaxes. Future work will include PostgreSQL and MySQL versions of BIRD to resolve these limitations and provide a more robust research environment for both NLP and DB experts.

## 9  Conclusion

In this paper, we introduce BIRD, a large-scale cross-domain, text-to-SQL benchmark with a particular focus on large database values. BIRD mitigates the gap between text-to-SQL research and real-world applications by exploring three additional challenges: 1) handling large and dirty database values, 2) external knowledge evidence, and 3) optimizing SQL execution efficiency. Our experimental results demonstrate that BIRD presents a more daunting challenge compared to existing benchmarks since even the most popular and powerful LLM, ChatGPT, falls significantly short of human performance. This leaves plenty of room for improvement and innovation in the text-to-SQL tasks. Moreover, our thorough efficiency and error analyses provide valuable insights and directions for future research, paving the way for the development of more advanced and practical text-to-SQL solutions in real-world scenarios.

## Acknowledgement

We thank all constructive comments from anonymous reviewers. Reynold Cheng, Jinyang Li, Ge Qu and Nan Huo were supported by the Hong Kong Jockey Club Charities Trust (Project 260920140) and the University of Hong Kong (Project 104006830). Chenhao Ma was supported by NSFC under Grant 62302421, Basic and Applied Basic Research Fund in Guangdong Province under Grant 2023A1515011280, Shenzhen Science and Technology Program ZDSYS20211021111415025. Jinyang Li and Ge Qu were supported by HKU Presidential PhD Scholar Programme. Ge Qu was also funded by Hong Kong PhD Fellowship Scheme. This work was supported by Alibaba Group through Alibaba Research Intern Program.

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

# A  Datasheet for Datasets

We follow instructions provided by `Datasheet for Datasets` to answer the important questions considering this dataset.

## A.1  Motivation

**For what purpose was the dataset created?**  The advancement of Large Language Models (LLMs) has raised concerns regarding whether state-of-the-art LLMs, such as ChatGPT and Codex, can replace human effort in real-world text-to-SQL tasks involving large database values. That is because their exceptional performance on previous academic tasks like SPIDER impresses researchers. However, we observe that current cross-domain text-to-SQL benchmarks only focus on the database schema, which lacks full attention to values, resulting in a gap between academic and real-world applications. To address this issue, we introduce BIRD, the largest cross-domain text-to-SQL benchmark highlighting extensive and realistic databases for community development. Additionally, we hope to observe the performance gap between LLMs and humans. Our experimental results indicate that, as of now, LLMs are still unable to replace human effort. As far as we know, BIRD is the first text-to-SQL benchmark to collect human performance.

**Who created the dataset (e.g., which team, research group) and on behalf of which entity (e.g., company, institution, organization)?**  Please refer to the author list for details. Our research team involves Star Lab at The University of Hong Kong, Alibaba DAMO Academy Conversational AI (ConAI) Team, the Department of Computer Science at the University of Illinois Urbana-Champaign, the Department of EECS at Massachusetts Institute of Technology, the School of Data Science at The Chinese University of Hong Kong (Shenzhen), and Database Group of Tsinghua University.

**Who funded the creation of the dataset?**  This dataset is fully funded by the Alibaba DAMO Academy ConAI team. We spent 97,654 USD for presenting this data. The budget includes 10% for recruiting competent research interns, 80% for developing the benchmark, and 10% for refining and implementing the benchmark.

## A.2  Composition

**What do the instances that comprise the dataset represent (e.g., documents, photos, people, countries)?**  BIRD contains natural language questions, external knowledge evidence sentences, processed large databases, database description files (csv), and SQL queries.

**How many instances are there in total (of each type, if appropriate)?**  BIRD contains 12,751 natural language questions, 12,751 external knowledge evidence sentences, 95 processed large databases, 95 folders of database description CSV files, and 12,751 ground truth SQL queries.

**Does the dataset contain all possible instances or is it a sample (not necessarily random) of instances from a larger set?**  In BIRD, we divide it into three sets: training, development, and testing. Training and development sets are public while testing data set is hidden for the fair evaluation of all text-to-SQL challengers. This could witness the real development of text-to-SQLs in the LLM era.

**Is there a label or target associated with each instance?**  In BIRD, we provide two labels for each question instance: SQLs (the target of input) and external knowledge evidence (expert annotated evidence for each expected SQL).

**Is any information missing from individual instances?**  No.

**Are relationships between individual instances made explicit (e.g., users' movie ratings, social network links)?**  No.

**Are there recommended data splits?**  Our data consists of 9,428 instances for the training set, 1,534 instances for the development set, and 1,789 instances for the concealed test set. The training

and development sets are derived from public databases, while the test set databases are curated and designed by our specialized team. We do this because some researchers express concerns that the remarkable performance of LLMs in text-to-SQL tasks may not be attributed to an improvement in capabilities, but rather to the exposure of data and database values to the LLMs during the pre-training phase. To address these concerns, we opt to self-design new databases in testing using actual tabular data, thereby ensuring that LLMs do not preview the databases.

**Are there any errors, sources of noise, or redundancies in the dataset?**    As stated in the main content, our double-blind annotation procedure is both expensive and rigorous, ensuring data quality. However, it is virtually impossible for any dataset, especially complex ones, to be entirely free of errors. Our team is committed to enhancing the data even after this paper is accepted, thereby contributing to the text-to-SQL community. In addition, we encourage users to provide feedback and report errors on our data website, allowing us to rectify and enhance the dataset.

**Is the dataset self-contained, or does it link to or otherwise rely on external resources (e.g., websites, tweets, other datasets)?**    Yes, all databases in training and development are collected under appropriate licenses. Please see Section 3.2 for more details

**Does the dataset contain data that might be considered confidential (e.g., data that is protected by legal privilege or by doctor-patient confidentiality, data that includes the content of individuals' non-public communications)?**    No.

**Does the dataset contain data that, if viewed directly, might be offensive, insulting, threatening, or might otherwise cause anxiety?**    No.

**Does the dataset identify any subpopulations (e.g., by age, gender)?**    Some questions mention ages and genders, but they are just used to detect the capability of models on text-to-SQLs. No bias or other opinions are involved.

**Is it possible to identify individuals (i.e., one or more natural persons), either directly or indirectly (i.e., in combination with other data) from the dataset?**    No. All databases are collected from open-sourced platforms, and any sensitive data has already been processed before.

**Does the dataset contain data that might be considered sensitive in any way (e.g., data that reveals race or ethnic origins, sexual orientations, religious beliefs, political opinions or union memberships, or locations; financial or health data; biometric or genetic data; forms of government identification, such as social security numbers; criminal history)?**    No, this is a QA-based text-to-SQL dataset, we don't require models to deliver any opinions on results. And also we don't present any bias or opinions in the dataset.

### A.3    Collection Process

**How was the data associated with each instance acquired?**    Section 3 and Appendix B.2 introduce this in detail.

**What mechanisms or procedures were used to collect the data (e.g., hardware apparatuses or sensors, manual human curation, software programs, software APIs)?**    Section 3 and Appendix B.2 introduce this in detail. Our crowdworkers use Alibaba internal labeling software to annotate the data and examine the results.

**If the dataset is a sample from a larger set, what was the sampling strategy (e.g., deterministic, probabilistic with specific sampling probabilities)?**    No.

**Who was involved in the data collection process (e.g., students, crowdworkers, contractors) and how were they compensated (e.g., how much were crowdworkers paid)?**    Four PhD students and two MS students are involved in the creation of database description files. Two independent teams of crowdworkers are recruited to annotate questions and SQLs. The question annotators are composed of 11 English native speakers and SQL annotators are comprised of database engineers and DB students. The total consumption is 97,654 USD.

**Over what timeframe was the data collected?**   From Sep. 2022 to Mar. 2023.

**Were any ethical review processes conducted (e.g., by an institutional review board)?**   Yes, we take such issues very seriously. During the review process, we found that certain questions related to politics or inappropriate language. We have addressed these concerns by modifying the content and providing a serious warning to the annotators responsible for such instances.

**Did you collect the data from the individuals in question directly, or obtain it via third parties or other sources (e.g., websites)?**   Section 3 and Appendix B.2 introduce this in detail.

**Were the individuals in question notified about the data collection?**   Yes.

**Did the individuals in question consent to the collection and use of their data?**   Sure, we recruited them and paid them satisfying salaries.

**If consent was obtained, were the consenting individuals provided with a mechanism to revoke their consent in the future or for certain uses?**   No.

**Has an analysis of the potential impact of the dataset and its use on data subjects (e.g., a data protection impact analysis) been conducted?**   Yes, we did a very comprehensive analysis including error analysis, and efficiency analysis, in the experiments of the paper and Appendix.

### A.4   Preprocessing/cleaning/labeling

**Was any preprocessing/cleaning/labeling of the data done (e.g., discretization or bucketing, tokenization, part-of-speech tagging, SIFT feature extraction, removal of instances, processing of missing values)?**   Yes, we provide the token list for each question and SQLs from NLTK for users.

**Was the "raw" data saved in addition to the preprocessed/cleaned/labeled data (e.g., to support unanticipated future uses)?**   No. Is the software that was used to preprocess/clean/label the data available? Yes, https://www.nltk.org/

### A.5   Uses

**Has the dataset been used for any tasks already?**   No.

**Is there a repository that links to any or all papers or systems that use the dataset?**   No.

**What (other) tasks could the dataset be used for?**   Sure, our databases and analysis-style questions are most valuable, so they could be beneficial to DB-based code generation, data science analysis, etc.

**Is there anything about the composition of the dataset or the way it was collected and preprocessed/cleaned/labeled that might impact future uses?**   No.

**Are there tasks for which the dataset should not be used?**   No.

### A.6   Distribution

**Will the dataset be distributed to third parties outside of the entity (e.g., company, institution, organization) on behalf of which the dataset was created?**   No.

**How will the dataset will be distributed (e.g., tarball on the website, API, GitHub)?**   All source codings and datasets could be found on our leaderboard website: https://bird-bench.github.io/. And we provide fast download links for the convenience of researchers who want to use our big data. Furthermore, the code repository can be found in https://github.com/AlibabaResearch/DAMO-ConvAI/tree/main/bird

**When will the dataset be distributed?** Now.

**Will the dataset be distributed under a copyright or other intellectual property (IP) license, and/or under applicable terms of use (ToU)?** Given the database size of BIRD is the largest until now, we are afraid that abusing ample database values may lead to inappropriate commercial use. Therefore, we claim that this dataset should be distributed under CC BY-NC 4.0.

**Have any third parties imposed IP-based or other restrictions on the data associated with the instances?** No.

**Do any export controls or other regulatory restrictions apply to the dataset or to individual instances?** No.

### A.7   Maintenance

**Who will be supporting/hosting/maintaining the dataset?** HKU STAR LAB and Alibaba DAMO Academy

**How can the owner/curator/manager of the dataset be contacted (e.g., email address)?** Contact bird.bench23@gmail.com or the corresponding authors or co-first authors in the author list.

**Is there an erratum?** No.

**Will the dataset be updated (e.g., to correct labeling errors, add new instances, delete instances)?** Yes, we will keep polishing and optimizing our data periodically.

**If the dataset relates to people, are there applicable limits on the retention of the data associated with the instances (e.g., was the individuals in question were told that their data would be retained for a fixed period of time and then deleted)?** No.

**Will older versions of the dataset continue to be supported/hosted/maintained?** No. The most updated version will be more reliable.

**If others want to extend/augment/build on/contribute to the dataset, is there a mechanism for them to do so?** Yes, but they should contact the authors first.

# B Appendix

## B.1 Text-to-SQL Difficulty

In order to help researchers deeply analyze model performance in various text-to-SQL case levels, we class all examples as `simple` (30%), `moderate` (60%), and `challenging` (10%). Previous work, such as SPIDER, computed difficulty mainly based on SQL complexity. However, we find that additional factors, such as question comprehension, schema linking, and external knowledge reasoning, also influence model and human performance. Therefore, each SQL annotator is required to evaluate examples based on these factors, and experts conclude the ratings to divide examples into the three aforementioned difficulty levels. This approach offers a more extensive difficulty analysis for text-to-SQL tasks. And the performance of ChatGPT on three different difficulty levels is shown in Table B.1. we take the approach of human scoring under established rules. A detailed crowdsourcing rule is employed to rate the difficulty when SQL annotators generate SQLs for each question. The process consists of evaluating four dimensions: The process consists of evaluating four dimensions:

1. **Question Understanding:** On a discrete scale from 1 to 3, annotators assess the ambiguity and difficulty of comprehending the question's intent, with 1 being straightforward, 2 being clear but requiring more thought, and 3 being extremely ambiguous.

2. **Knowledge Reasoning:** On a discrete scale from 1 to 3, annotators rate the amount of external knowledge required to map the question to SQL, with 1 indicating no knowledge is required, 2 requiring evidence of external knowledge for generating SQLs that is easy to understand, and 3 requiring extensive knowledge and much more thoughts.

3. **Data Complexity:** Annotators rate the complexity of schema relations and data size that need analyzing on a discrete scale of 1-3, with 1 being a simple schema and data, 2 being complex schema and values understandable through database description files, and 3 being highly complex and difficult to comprehend values and schema even with description files.

4. **SQL Complexity:** Annotators rate the syntactic complexity of the target SQL query on a discrete scale of 1-3, with 1 being a simple SQL without many keywords, 2 being more complicated than 1, and 3 being a highly complex SQL with many functions and

Each dimension is considered equally important for text-to-SQL annotations. SQLs are ranked based on these scores, and we present simple, moderate, and challenging difficulties at proportions of 30%, 60%, and 10%, respectively.

| MODEL | DEV SET | | | | TEST SET | | | |
|---|---|---|---|---|---|---|---|---|
| | simple | moderate | challenging | total | simple | moderate | challenging | total |
| (EX) ChatGPT | 31.08 | 13.29 | 12.08 | 24.05 | 35.41 | 19.46 | 12.28 | 26.77 |
| (EX) ChatGPT + KG | **45.44** | **26.14** | **19.01** | **37.22** | **49.21** | **31.89** | **20.70** | **39.30** |
| (VES) ChatGPT | 36.20 | 15.43 | 14.42 | 27.97 | 50.09 | 24.71 | 15.39 | 36.68 |
| (VES) ChatGPT + KG | **54.71** | **28.16** | **22.80** | **43.81** | **65.06** | **41.21** | **25.81** | **51.40** |

Table 4: The Execution Accuracy (EX) and Valid Efficiency Score (VES) are presented for both the ChatGPT model and its version with grounding (KG) for external knowledge evidence, taking into consideration development and testing datasets.

## B.2 Annotation Entrance

**Annotation Platform and Compensation.** The data is collected from **Alibaba-Appen**[§], an internal version. Each Question annotator receives a $0.6 reward for each validated question, while SQL annotators earn $1 per SQL contribution. We also invite text-to-SQL experts and professors to join to check and annotate external knowledge evidence without compensation. There are ~1340 SQLs confirmed per week.

---

[§]https://appen.com/crowd-2/#crowd

**Text-to-SQL Experts.** The three full-time text-to-SQL experts in this project are: (1). A database research scientist who's published over 20 top DB conference papers (e.g., SIGMOD, VLDB). (2). A PhD student with research interests in text-to-SQL, who achieved state-of-the-art results on text-to-SQL open challenges. (3). A DBA engineer with more than 10 years of experience in text-to-SQL applications for both B2B and B2C businesses.

**Question Annotation Entrance.** We hire a group of native speakers of English with degrees above the bachelor's level and database-related knowledge to ask a variety of natural language questions regarding the values of databases. To fulfill this objective, we have adopted the following procedure: (1). ER diagrams and database description files are documented to assist the annotators in understanding the databases; (2). we present the annotators with three databases from different domains and require them to generate 10 questions for each database; (3). these questions are then assessed by 3 text-to-SQL experts applying predefined rules. Those questions earning at least two votes are marked as valid. Only annotators capable of generating no less than 8 valid questions per database are preserved. As a result, 11 native speakers contribute questions to BIRD.

**SQL Annotation Entrance.** With the purpose of enhancing the quality of our SQL queries, we assemble a team of skilled data engineers and database students. The team undergoes rigorous testing through the text-to-SQL evaluation process, which assesses their capability of generating SQL queries for a variety of questions facing different domains of databases. Each annotator is asked to answer 10 questions, and only those who score at least 9 out of 10 will be qualified to annotate SQL queries for BIRD.

### B.3 Question Distribution

Figure 8 contains the detailed question types and their examples.

### B.4 Experiment Details

**FT-based Models.** T5 is a strong and versatile pre-trained language model (PLM) for text-to-text generation that has achieved state-of-the-art performance in a variety of semantic parsing tasks, including text-to-SQL. We concatenate the question with serialized database schema as input [40, 49, 41]. And SQL can be fetched in an end-to-end fashion by easily fine-tuning. While seq2AST-based methods [43, 5] are also effective in text-to-SQL, actually their grammar rules utilized during decoding are constrained on specific datasets [25]. We implement our codes mainly based on the hugging-face transformers library [¶]. We set the max input length as 1024, the generation max length as 512, and the batch size as 32. We also adopt Adafactor as our primary optimizer with a linear decayed learning rate of 5e-5. All experiments are conducted on one NVIDIA Tesla A100 80GB, which is available for most research centers. We set the random seed as 1 for all runs of FT-based models since 1 is an optimal seed proven by previous SOTA models [27, 49].

**ICL-based Models.** Codex (`code-davinci-002`) and ChatGPT (`gpt-3.5-turbo`) are popular and powerful large-scale pre-trained language models (LLMs) for code generation driven by ICL. They can produce multiple types of codes, including SQL, from human instructions without additional training. We employ programming-based prompts, as described in [39], to collect results by calling the API. Also, we choose the Azure OpenAI API to align the codes with other variants of LLMs. Given that models are not allowed access to unseen databases and ground-truth SQLs in the evaluation set, a zero-shot generation strategy is the most appropriate. Moreover, to investigate the impact of multi-step reasoning of LLMs on BIRD, we implement the Chain-Of-Thought (COT) technique [48] by easily adding the prompt sentence `"Let's think step by step."` before the generation of SQLs [21]. However, we find out the output of ChatGPT is too uncertain with many unexpected explanations, thus we provide a 1-shot pseudo example for ChatGPT to learn the procedure of thinking and output format. The detailed prompt design is shown in Figure 9. In order to minimize the randomness of results, we set the temperature as 0 to ensure reproduction.

**Knowledge Fusion.** In the baseline implementation, we naively concatenate the knowledge evidence sentences with questions and database schemas, but we can observe a significant improvement

---

[¶] https://huggingface.co/

| Question Type | Sub Type | Question / SQL | Percentage |
|---|---|---|---|
| Fundamental Type | Match-based | How many gas stations in CZE has Premium gas? 

 `SELECT COUNT(GasStationID) FROM gasstations WHERE Country = 'CZE' AND Segment = 'Premium'` | 83.9 % |
| | Ranking | What are the titles of the top 5 posts with the highest popularity? 

 `SELECT Title FROM posts ORDER BY ViewCount DESC LIMIT 5` | 20.3 % |
| | Comparison | How many color cards with no borders have been ranked higher than 12000 on EDHRec? 

 `SELECT COUNT(id) FROM cards WHERE edhrecRank > 12000 AND borderColor = 'borderless'` | 16.7 % |
| | Counting | How many of the members' hometowns are from Maryland state? 

 `SELECT COUNT(T2.member_id) FROM zip_code AS T1 INNER JOIN member AS T2 ON T1.zip_code = T2.zip WHERE T1.state = 'Maryland'` | 30.4 % |
| | Aggregation | What is the average height of the superheroes from Marvel Comics? 

 `SELECT AVG(T1.height_cm) FROM superhero AS T1 INNER JOIN publisher AS T2 ON T1.publisher_id = T2.id WHERE T2.publisher_name = 'Marvel Comics'` | 15.7 % |
| Reasoning Type | Domain Knowledge | Name the ID and age of patient with two or more laboratory examinations which show their hematoclit level exceeded the normal range. 

 `SELECT T1.ID, STRFTIME('%Y', CURRENT_TIMESTAMP) - STRFTIME('%Y', T1.Birthday) FROM Patient AS T1 INNER JOIN Laboratory AS T2 ON T1.ID = T2.ID WHERE T1.ID IN ( SELECT ID FROM Laboratory WHERE HCT > 52 GROUP BY ID HAVING COUNT(ID) >= 2 )` | 23.6 % |
| | Numeric Computation | Among the posts with a score of over 20, what is the percentage of them being owned by an elder user? 

 `SELECT CAST(SUM(IIF(T2.Age > 65, 1, 0)) AS REAL) * 100 / count(T1.Id) FROM posts AS T1 INNER JOIN users AS T2 ON T1.OwnerUserId = T2.Id WHERE T1.Score > 20` | 24.5 % |
| | Synonym | How many clients opened their accounts in Jesenik branch were women ? (female) 

 `SELECT COUNT(T1.client_id) FROM client AS T1 INNER JOIN district AS T2 ON T1.district_id = T2.district_id WHERE T1.gender = 'F' AND T2.A2 = 'Jesenik'` | 7.2 % |
| | Value Illustration | Among the weekly issuance accounts, how many have a loan of under 200000? 

 `SELECT COUNT(T1.account_id) FROM loan AS T1 INNER JOIN account AS T2 ON T1.account_id = T2.account_id WHERE T2.frequency = 'POPLATEK TYDNE' AND T1.amount < 200000` | 70.1 % |

Figure 8: Questions in the BIRD contain two main categories. The `Fundamental Type` of questions are comparable to other text-to-SQL benchmarks. The `Reasoning Type` of questions requires external knowledge grounding to answer.

by this easy method. A more complicated and effective strategy of knowledge grounding for ChatGPT and T5 would be an important future topic. The knowledge evidence sentences are concluded to the external knowledge provided by annotators as described in Section 3.3.

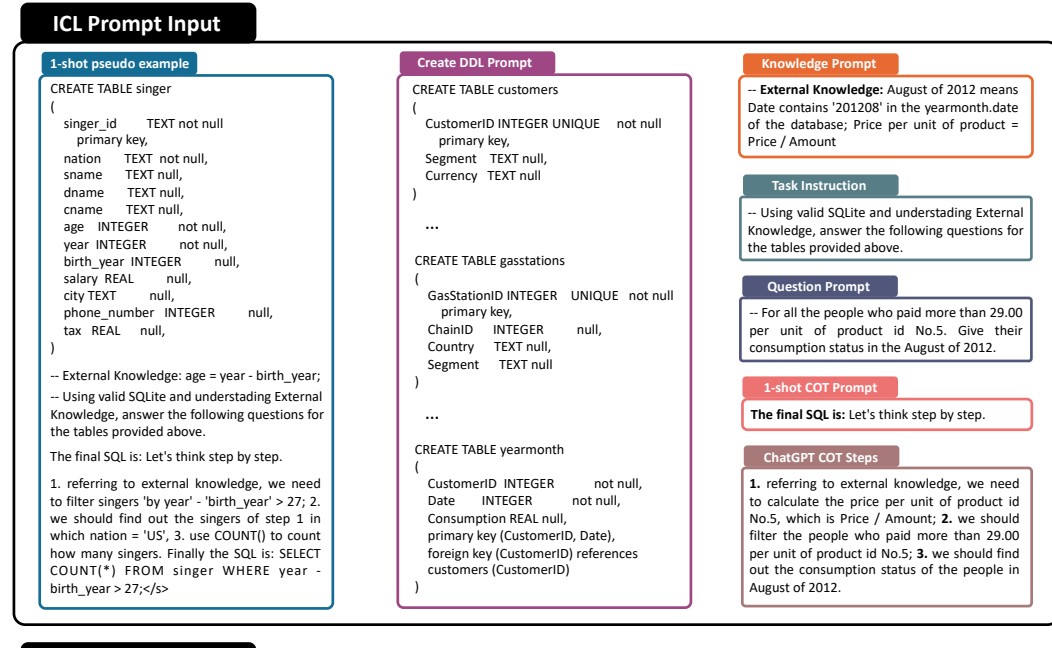

Figure 9: The detailed prompt design for implementation of ChatGPT + KG + COT.

## B.5 Efficiency Analysis Details

Two strategies for performing text-to-efficient-SQL are presented in Figure 10. Examples show that both two-stage optimization and embodied databases can help semantic parsings generate more efficient SQLs.

## B.6 Error Analysis Details

Figure 11 presents a detailed analysis of errors made by ChatGPT.

## B.7 Evaluation Details

During double-blind annotation in BIRD, we encountered numerous ambiguous issues that led to mismatches, predominantly due to unclear user intents. The most serious ambiguity is the use of "DISTINCT". Some annotators believe it should present only unique values, such as names, and cities, while others argue that it should be used only when questions explicitly mention "different" or "distinctive". Therefore, we use HashSet rather than List to compare final results since HashSet disregards row order and automatically filters repetitive rows to reduce this ambiguity. However, this may result in false positives for questions utilizing "ORDER BY." We identify three "ORDER BY" usage scenarios in BIRD: **1) Rank-based questions** (e.g., "Show me the top 5 students according to their math scores"): The order is less important as long as the results contain the correct students. **2) Superlative questions**: (e.g., "List the longest river in the USA"): The answer typically contains only one item (or tied results), so the impact is minimal. **3) Questions requiring a specific order** (e.g., "Show me the top five students based on their math scores in descending order"): This scenario explicitly requires correct ordering and may lead to false positives. However, such instances are uncommon, accounting for less than 1% of BIRD.

---

**Query Rewriting**

---

**Ex1.1 Question:**
List out the age of users who located in Vienna, Austria obtained the badge?

**ChatGPT SQL:**
```
SELECT Age FROM users WHERE Location = 'Vienna, Austria' AND Id IN (SELECT UserId FROM badges)
```

**Optimized SQL:** (time-saving percentage: **99.92%**)
```
SELECT u.Age FROM users AS u INNER JOIN badges AS b ON u.Id = b.UserId WHERE u.Location = 'Vienna, Austria'
```

**Take Away:**
By applying a **JOIN** operation instead of a subquery with **IN** can improve efficiency, as the database may execute the **JOIN** and filtering processes concurrently in just one operation **without** the need to store the intermediate results to filter primary query.

---

**Ex1.2 Question:**
How many of the members' hometowns are from Maryland state?

**ChatGPT SQL:**
```
SELECT COUNT(*) FROM member INNER JOIN zip_code ON member.zip = zip_code.zip_code WHERE zip_code.state = 'Maryland'
```

**Optimized SQL:** (time-saving percentage: **67.93%**)
```
SELECT COUNT(member.member_id) FROM member INNER JOIN zip_code ON member.zip = zip_code.zip_code WHERE zip_code.state = 'Maryland'
```

**Take Away:**
Utilizing the **COUNT** function on a **NOT-NULL** column, as opposed to **COUNT(*)**, can increase time efficiency. This rewritten SQL enables the database to count **NOT-NULL** values within a single column, rather than compute all rows including those with **NULL** values. Usually, the primary key column is selected as this **NOT-NULL** column.

---

**Ex1.3 Question:**
Who is the owner of the account with the largest loan amount?

**ChatGPT SQL:**
```
SELECT c.client_id FROM client c INNER JOIN disp d ON c.client_id = d.client_id INNER JOIN loan l ON d.account_id = l.account_id ORDER BY l.amount DESC LIMIT 1
```

**Optimized SQL:** (time-saving percentage: **62.39%**)
```
SELECT c.client_id FROM client c INNER JOIN disp d ON c.client_id = d.client_id INNER JOIN loan l ON d.account_id = l.account_id WHERE l.amount = ( SELECT MAX(amount) FROM loan)
```

**Take Away:**
In an unindexed environment, employing the **MAX** function can potentially yield faster results since it avoids the need for **sorting**, which could run against a large table.

---

**Adding Indexes to Database**

---

**Ex2.1 Question:**
How many accounts are there in the district of \"Pisek\"?

**ChatGPT SQL:**
```
SELECT COUNT(*) FROM account a INNER JOIN district d ON a.district_id = d.district_id WHERE d.A2 = 'Pisek'
```

**Added Indexes:** (time-saving percentage: **87.27%**)
```
CREATE INDEX account_district_id_index ON account(district_id);
CREATE UNIQUE INDEX district_district_id_uindex ON district(district_id);
```

**Take Away:**
Adding **indexes** into a database can significantly increase the speed of SQL queries because it creates a data structure that enables the database engine to quickly locate rows that match specific criteria instead of **scanning** the entire table.

---

Figure 10: Two possible solutions and explanations to improve efficiency are presented. The first batch of examples shows how to optimize SQL efficiency by rewriting SQL based on rules. The last example is to show that adding indexes to databases can also improve SQL efficiency without rewriting them.

## B.8 VES Details

Regarding $E$, in our experiment, we consider time as the main metric to represent efficiency, where $E \in (\epsilon, 30s)$. Here, $\epsilon$ is a small positive constant to prevent floating-point overflow. The single $E$ is not stable due to machine status. Lower $E$ refers to faster execution speed, which is more efficient.

Concerning $R$, it represents a normalized efficiency ratio between human-annotated SQL queries and predicted SQL queries to reduce the influence of machine status. The stability of this metric is ensured by running this computation 100 times for each example, filtering outliers, and subsequently

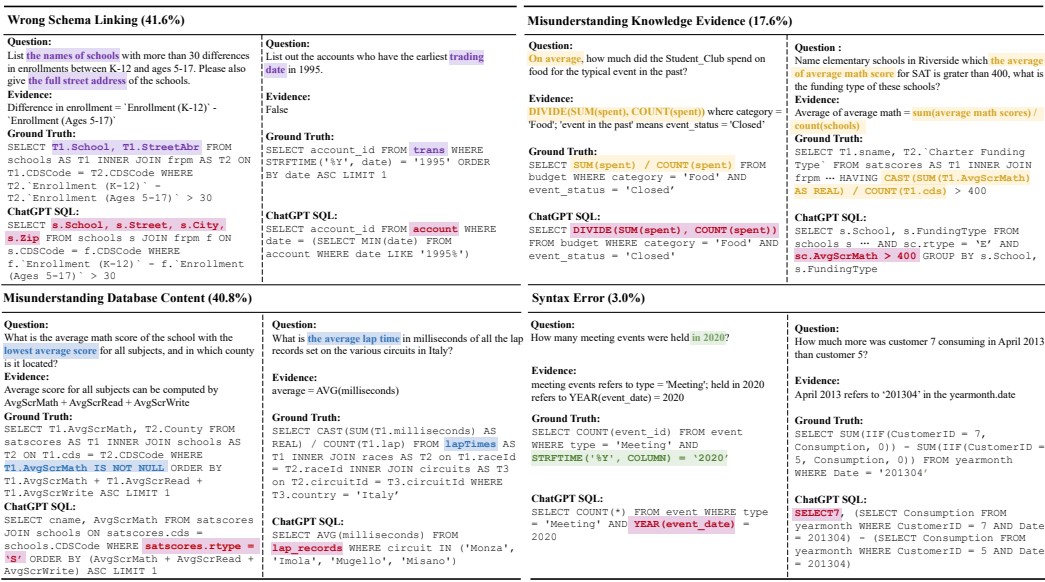

Figure 11: 4 major types of error cases are presented. Some cases are shortcuts for better presentation.

computing the average. Considering the rapid advancement of technology, it is impractical to anticipate the fastest SQL performance. Currently, the range for the efficiency ratio, $R$, is defined as $R \in (0, +\infty)$. $E(\hat{Y}_n)$ (efficiency of predicted SQL) is much lower than $E(\hat{Y})$ (efficiency of ground truth SQL according to EX), then the relative efficiency score $R$ will be increased. In short, higher $R$ refers to higher efficiency.

When measuring VES, we run 100 times for each SQL in the same CPU and evaluate average results after dropping the outliers. The STD of VES on dev set and test set after 10 trials are 0.043 and 0.025 respectively. We detect outliers in the following procedures:

1. Compute the mean and standard deviation of the dataset.

2. Then calculate the lower threshold as mean $- 3 \times$ standard_deviation and the upper threshold as mean $+ 3 \times$ standard_deviation.

3. Statistically, approximately 99.7% of the data points fall within 3 standard deviations of the mean.

## B.9 Human Performance Collection

The procedure of collecting human performance is still rigorous. During the annotation, all data is divided into 10 batches for better management and error tracks by experts. The first 8 batches of data are the final training data and dev data for public use, and the remaining 2 batches of data are used for testing. We consider the annotation of the first 8 batches of data as a learning process for SQL annotators since their erroneous SQLs could be fixed by experts and learn how to generate good-quality SQLs for this task. Then their first scores on an examination, conducted by testing set from the final two batches can be viewed as human performance since we don't interrupt and assist them during the examination and all errors are preserved. After testing, we proceed with the following double-blind SQL annotation procedures as Section 3.4 to correct SQLs for these data by a discussion with experts. And SQLs after the second round of double-blind annotation are collected as ground truth.

## B.10 Distribution of Open-source Databases

The databases in BIRD are all in accordance with one of following licenses:

**Public Domain**   Public Domain Mark
A public domain license refers to a legal designation that allows intellectual property, such as creative works or inventions, to be freely used, shared, and built upon by anyone without restrictions. When a work is in the public domain, it is no longer protected by copyright, patent, or trademark laws.

**CC-BY**   Creative Commons Attribution 4.0 International
This license is one of the open Creative Commons licenses and allows users to share and adapt the dataset so long as they give credit to the creator.

**CC-BY-SA**   Creative Commons Attribution-ShareAlike 4.0 International
This license is one of the open Creative Commons licenses and allows users to share and adapt the dataset so long as they give credit to the creator and distribute any additions, transformations, or changes to the dataset under this license.

**GPL**   General Public License
The GPL was created by the Free Software Foundation (FSF) and is also known as the GNU GPL, as it is used by the GNU Project. And it allows users to use, study, share, and modify the software under certain terms and conditions.

**CPOL**   Code Project Open License
It is a software license that is often used for articles, tutorials, and sample code shared on The Code Project website. The CPOL is intended to be a more permissive license, allowing developers to use, modify, and distribute the software without many of the restrictions imposed by other licenses like the GPL.

**CC0**   Creative Commons Zero
It is a public domain dedication tool created by Creative Commons. It allows creators to waive all their copyright and related rights in a work, effectively placing it in the public domain. This means that anyone can freely use, share, modify, and build upon the work without seeking permission or providing attribution to the original creator.

## B.11   SQL Function Taxonomy

SQL functions in BIRD mentioned in Table 1 span across multiple categories including:

- Window Functions, i.e., `OVER()`

- Date Functions, i.e., `JULIANDAY()`

- Conversion Functions, i.e., `CAST()`

- Math Functions, i.e., `ROUND()`

- String Functions, i.e., `SUBSTR()`

## B.12   Keyword Statistic

We have conducted a comprehensive analysis of the keywords employed in the BIRD dataset and visualize the results in the form of a nice-looking word cloud, which can be found in Figure 12. We further classify keywords into 7 following categories:

**Main Body Keywords**   • `SELECT` • `FROM` • `WHERE` • `AND` • `OR` • `NOT` • `IN` • `EXISTS` • `IS` • `NULL` • `IIF` • `CASE` • `CASE WHEN`.

**Join Keywords**   • `INNER JOIN` • `LEFT JOIN` • `ON` • `AS`.

**Clause Keywords**   • `BETWEEN` • `LIKE` • `LIMIT` • `ORDER BY` • `ASC` • `DESC` • `GROUP BY` • `HAVING` • `UNION` • `ALL` • `EXCEPT` • `PARTITION BY`.

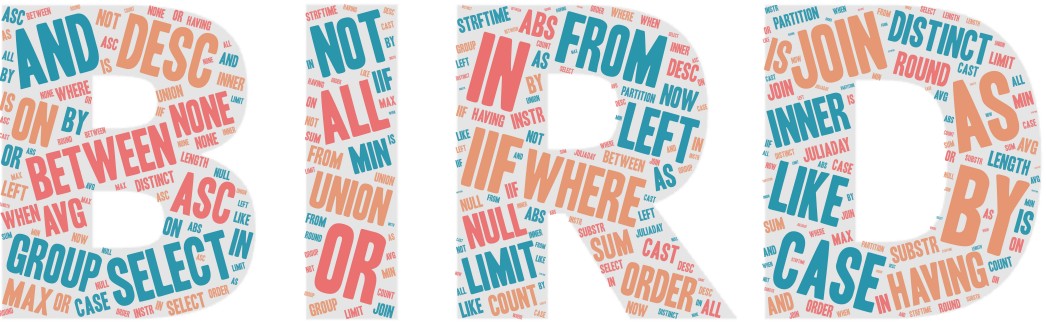

Figure 12: Keyword cloud presentation for SQLs in BIRD.

**Aggregation Keywords** • AVG • COUNT • MAX • MIN • ROUND • SUM.

**Scalar Keywords** • ABS • LENGTH • STRFTIME • JULIADAY • NOW • CAST • SUBSTR • INSTR.

**Comparison Keywords** • = • > • < • >= • <= • !=.

**Computing Keywords** • − • + • * • /.

### B.13  Study about Text-to-SQL Models

The fundamental principle of a cross-domain text-to-SQL parser involves the construction of an encoder to learn representations of questions and schemas, followed by a decoder to generate SQLs [37]. For example, IRNET [12] designs an encoder consisting of attention-based Bi-LSTM for learning question and schema representations, and a decoder to predict SQLs based on the encoded intermediate representations. RATSQL [43], SDSQL [17], LGESQL [5], and S$^2$SQL [18], Proton [44] enhance the representation learning of natural language questions and database schema via relational graph neural network. R$^2$SQL [16], SCoRe [54], and STAR [4] enhance contextual learning for conversational text-to-SQL tasks. Later, sequence-to-sequence pre-trained language models (PLMs) such as T5 [38] become popular in text-to-SQL tasks due to their portability and capability of generation across different datasets. These models achieve impressive results by fine-tuning with minimal effort. Furthermore, RASAT [36] enhances T5's structural information encoding via schema alignment into the encoder, while Graphix [27] equips T5 with multi-hop reasoning to achieve state-of-the-art results on complicated cross-domain text-to-SQL tasks. In recent years, LLMs such as ChatGPT [33], Palm [8], OPT [56], have attracted considerable attention due to their powerful zero-shot reasoning and domain generalization capabilities. ChatGPT can perform exceptionally well on semantic parsing tasks, including text-to-SQL tasks, with minimal input data. In fact, in the BIRD project, ChatGPT even performs more impressively than initially expected.

**Study about SQL Efficiency**  Efficient execution of SQL queries on big databases has been a significant topic in both academia and industries. Many techniques are proposed to improve SQL query efficiency, by index selection [22], SQL optimization [26, 61], etc. SQL optimization is a common method for enhancing the efficiency of SQL queries. Several SQL optimization algorithms [28, 30, 47], such as rule-based optimization and cost-based optimization, are proven effective. Rule-based optimization employs a set of principles to transform the SQL query into a form that can be executed more efficiently. On the other hand, cost-based optimization estimates the execution cost of various query plans and selects the one with the lowest cost by analyzing the statistic distribution of database values. Similar to the NLP community, there are also recent works utilizing artificial intelligence for query optimization such as [61]. Index prediction is another important technique for improving SQL execution efficiency. Researchers propose many algorithms of index prediction [60] based on various optimization criteria, such as minimizing SQL execution time, and maximizing index utilization. In this work, we provide VES to measure the efficiency of text-to-SQL generators to encourage them to generate accurate and fast SQLs for users.

