# OpenReview forum: "Can LLM Already Serve as A Database Interface? A BIg Bench for Large-Scale Database Grounded Text-to-SQLs"
_NeurIPS.cc/2023/Track/Datasets_and_Benchmarks — NeurIPS 2023 Datasets and Benchmarks Spotlight_

### Official Review · Reviewer_7VcG · 2023-07-17
**Can LLM Already Serve as A Database Interface? A BIg Bench for Large-Scale Database Grounded Text-to-SQLs**

**Rating:** 9
**Confidence:** 5
**Correctness:** Yes.
**Clarity:** Yes.

**Strengths:**

- A new large-scale collection of realistic text-to-SQL dataset
- Incorporating external knowledge paired with the database
- Incorporating SQL efficiency for the first time in text-to-SQL evaluation
- Clearly written paper
- Data generation process is easy to follow

**Additional Feedback:**

No.

**Documentation:**

Yes.

**Ethics:**

No.

**Limitations:**

- As incorporating external knowledge evidence is one the core contributions, is there an ablation study that illustrates how each of four categories (Domain Knowledge, Numeric Computing, Synonym, Value Illustration) affect the performance improvement in models?
- How is external knowledge consumed in T5 models during training?
- Is there a fixed rule for annotating SQL difficulty ratings?

**Opportunities For Improvement:**

- Minor typos (line 195: tex-to-SQL)

**Relation To Prior Work:**

Yes.

**Summary And Contributions:**

This paper proposes a novel dataset called BIRD, which aims to bridge the gap between academic study and real-world applications. The key aspects of the dataset are 1) creating databases containing a large number of rows, 2) having external knowledge for better grounding between NL questions and database values, and 3) proposing SQL efficiency evaluation.

The authors collected numerous open-source relational databases from real analysis platforms to incorporate realistic databases. They also gathered natural language questions and their corresponding SQL queries by crowdsourcing. The experiments show that LLMs such as ChatGPT and GPT-4 perform significantly worse than humans on the dataset.

---

> ### Author Response · Authors · 2023-08-20
> **Thanks for your appreciation and comments**
>
> Thank you for your appreciation and valuable evaluation of our work.
>
> **Q1 : Is an Ablation study that illustrates how each of four categories (Domain Knowledge, Numeric Computing, Synonym, Value Illustration) affect the performance improvement in models?**
>
> Thanks for your insighful suggestion, and please refer to **General Response Q3**.
>
> **Q2 : How is external knowledge consumed in T5 models during training?**
>
> In the T5 model, we simply concatenate external knowledge, schema items (table, column names), and the external knowledge evidence sentence as [1], which is a mainstream and effective method for text-to-text PLM for consuming additional information. We believe that future work can develop more complicated methods to improve the integration of external knowledge.
>
> [1] Xie et al., UnifiedSKG: Unifying and Multi-Tasking Structured Knowledge Grounding with Text-to-Text Language Models.
>
>
> **Q3 : Is there a fixed rule for annotating SQL difficulty ratings?**
>
> Yes, we take the approach of human scoring under established rules. A detailed crowdsourcing rule is employed to rate the difficulty when SQL annotators generate SQLs for each question. The process consists of evaluating four dimensions:
>
> - **Question Understanding:** On a discrete scale from 1 to 3, annotators assess the ambiguity and difficulty of comprehending the question's intent, with 1 being straightforward, 2 being clear but requiring more thought, and 3 being extremely ambiguous.
>
> - **Knowledge Reasoning:** On a discrete scale from 1 to 3, annotators rate the amount of external knowledge required to map the question to SQL, with 1 indicating no knowledge is required, 2 requiring evidence of external knowledge for generating SQLs that is easy to understand, and 3 requiring extensive knowledge and much more thoughts.
>
> - **Data Complexity:** Annotators rate the complexity of schema relations and data size that need analyzing on a discrete scale of 1-3, with 1 being a simple schema and data, 2 being complex schema and values understandable through database description files, and 3 being highly complex and difficult to comprehend values and schema even with description files.
>
> - **SQL Complexity:** Annotators rate the syntactic complexity of the target SQL query on a discrete scale of 1-3, with 1 being a simple SQL without many keywords, 2 being more complicated than 1, and 3 being a highly complex SQL with many functions and
>
> Each dimension is considered equally important for text-to-SQL annotations. SQLs are ranked based on these scores, and we present simple, moderate, and challenging difficulties at proportions of 30%, 60%, and 10%, respectively.

---

> > ### Comment · Reviewer_7VcG · 2023-08-29
> > **Official Comment by Reviewer 7VcG**
> >
> > I appreciate the time and effort the authors put into their response. I raised my score.

---

> > > ### Author Response · Authors · 2023-08-30
> > > **Thank you for your review**
> > >
> > > Thanks for your insightful suggestions and appreciation of our work!

---

### Official Review · Reviewer_fFFH · 2023-07-21
**Review submission 447**

**Rating:** 8
**Confidence:** 3

**Strengths:**

- The submission introduces BIRD, an extensive benchmarking dataset to evaluate text-to-SQL tasks which includes hard datasets and requires reasoning to include external knowledge on the DB to produce an optimal result.

- The Valid Efficiency Score (VES) metric is introduced as a means of evaluating not only the correctness of the query result provided by the LLM, but also whether or not the query is efficient according to some metric (e.g. the execution time).

- By emphasising the need to reason and understand external information, the benchmark can model a more realistic scenario for how a human would approach the problem of creating a SQL query on a given Database. This makes the benchmark more realistic and highlights how LLMs need to be able to understand external information to work well.

- The experimental section provides good insight in the shortcomings of one of the methods under evaluation (ChatGPT), suggesting potential avenues of improvement.

**Additional Feedback:**

- In tables 2 and 3, the position of `FT-based` and `ICL-based` should be left-aligned with the values in the `Models` column. `Human Performance` should use the same font as the two other headers.

- In section 4, it would be nice to have some examples of the `Fundamental type` questions to go along with the `Reasoning type` examples in the additional material.

**Clarity:**

The paper is often unclear and hard to follow.

- In section 2, what does the single parameter $\theta$ mean? The entire collection of parameters required to train the LLM?

- In section 3.3, the 8 math operations are split over the main body and a footnote, when they could all be listed in the same location.

**Correctness:**

The dataset provided as contribution has been constructed in a sound way.

The experiments provided with the baseline are correct, but somewhat limited in that only one of the LLMs was evaluated in detail (ChatGPT). Additional experiments should include
- The cost of querying the LLM as part of the metrics used in the VES.
- A metric that tracks the number of LLM-generated queries that required human intervention to run, as well as the amount of effort (e.g., difference in the number of tokens) required to get a query to work.

**Documentation:**

The website and documentation provided with the submission are satisfactory.

**Ethics:**

As a major part of the work involves crowdsourcing, participants should have received appropriate compensation for the contribution.

LLMs are notorious for their propensity to allucinate incorrect information. This is critical in production environments since blindly relying on the output of a LLM can lead to issues. While the results provided in this work suggest that the current solutions are still far from being reliable enough to be used in a wholly unsupervised fashion (with no expert vetting the results), there should be a discussion on the risk of using the current generation of LLMs with no supervision. Furthermore, the problem of LLMs producing incorrect code is almost "handwaved away" in section 3.4, where it is mentioned that incorrect queries are corrected by experts and then evaluated.

**Limitations:**

- In section 3.4 it is stated that "if executed result set is "NULL", experts will make *slight* changes [...] until the associated SQLs can provide a valid result set", then further on "if the evidence is insufficient to generate SQL or contains errors, experts will be in charge of correcting them". This is a critical lack of detail on the kind of corrections that were required and the amount of effort needed to fix the LLM-generated SQL in order to get them to work. If the expert-corrected queries were considered correct for the EX and VES scores (as the result of the query was correct), then they should still be penalized in some way due to the fact that human intervention was required to make them viable. There should be some figure on how often this pattern occurs, and partially correct queries should also be kept track of and tallied when evaluating the performance.

- As querying LLMs incurs in a non-negligible cost (as is also pointed out by the authors, e.g., in section 6.4), the query cost/cost per token should also be accounted for when selecting the most suitable LLM, as it would be a relevant metric to rank models by.

**Opportunities For Improvement:**

- It is not clear why the datasets in BIRD are manually split over train/test/dev, rather than providing the full set of datasets as is and letting the user split them as they seem fit, e.g., for cross-validation purposes.  A better justification for this division should be provided.

- In the paragraph on SQL statistics, it is not clear how the number of tokens (what strategy was used?). This should be explained.

- The definitions of EX and VES are quite unclear. EX would benefit from an example. For VES, the definition of "efficiency" should be at the start of the paragraph.

- In Figure 2, each step introduces new terms to describe the crowdsourced experts that are involved ("specialists", "experts", "question annotators", "SQL annotators"). It is never clear who is actually involved in these steps and whether there is overlap between the different categories. Section 3.4 does not provide a definition for the various "classes" either, so it is very difficult to match the roles.

**Relation To Prior Work:**

Prior work is accounted for and the submission clearly highlights its difference from previous contributions.

**Summary And Contributions:**

The submission introduces BIRD, a BIg bench for laRge-scale Database, whose objective is benchmarking the performance of text-to-SQL tasks. In Text-to-SQL, the objective is producing a query over a Database by feeding a model (typically a LLM)  a set of instructions as free text.

This problem is extremely relevant in various fields, from academia to the industry, due to its potential to allow non-technical users to produce complex queries over databases without having to know a query language. The introduction of LLMs such as ChatGPT has kickstarted this subject as it is now possible to produce working code from a set of text instructions.

However, current-generation LLMs are not perfect, as the evaluation carried out in this submission sets out to prove by building a new benchmarking dataset. BIRD is a large scale collection of DBs sourced from publicly available data, paired with a collection of SQL queries crafted by human experts. This combination allows to ask LLMs to produce queries on a given DB, then evaluate whether the produced query is correct or not by comparing it with the best human-made solution.

There is a large emphasis on the need to understand the DB. This is achieved via External Knowledge, which includes information on the domain and value mappings among others. Modeling external knowledge fundamental to achieve good results in a real scenario, which makes its inclusion in BIRD an important contribution.

The evaluation is carried out by using two efficiency metrics in Execution Accuracy (EX) and Valid Efficiency Score (VES). The first evaluates whether the result of the LLM-generated queries is correct, while the second weighs EX by additional metrics to measure the efficiency of the LLM solution compared to the human solution. VES allows to account for the potential lower efficiency of a LLM-generated query when compared to a human-generated query.

The experimental results show that BIRD is a hard benchmark, as the gap between LLMs and human performance is significantly larger than previous benchmarks. The results also give good insight in the main shortcomings of LLMs and thus suggests potential future work, which should be made easier thanks to the open release of the benchmark.

---

> ### Author Response · Authors · 2023-08-20
> **Thanks for your suggestions and reviews**
>
> Thank you for your kind feedback and valuable comments. We have addressed your concerns by answering the following questions:
>
> **Q1: Why the datasets in BIRD are manually split over train/test/dev?**
>
> BIRD is specifically designed to evaluate the ability of models to generalize to new and unseen databases. To achieve this goal, we split the datasets in such a way so that the testing databases are completely hidden from the training process. Rather than relying on the models' ability to memorize databases encountered during training, this method permits a more accurate evaluation of their capability of generating correct SQLs in real scenarios.
>
> In addition, training text-to-SQL models is a resource-intensive process. The T5-large model, for instance, requires four days to attain convergence. Therefore, conducting cross-validation in this context becomes impractical and time-consuming for users. This emphasizes the importance of a well-designed dataset division for evaluating the generalization capabilities of models efficiently and effectively. Above all, we used a cross-domain setting with a hidden test set to provide a fair evaluation.
>
> **Q2: How does the number of SQL tokens?**
>
> We directly use the `word_tokenize` from the package `nltk` to tokenize the SQLs and then compute the number of tokens.
>
> **Q3: The definitions of EX and VES are quite unclear. EX would benefit from an example. For VES, the definition of "efficiency" should be at the start of the paragraph.**
>
> As stated from the line 163 to 180, EX stands for **Execution Accuracy**. It measures the percentage of predicted SQL queries that produce the same result as the ground truth SQL query when executed on the database. For example, if there are 100 text-to-SQL examples and the model's predicted SQL queries produce the same results as the ground truth for 70 of them, the EX would be calculated as 70/100, or 70%.
> On the other hand, VES, i.e., **Valid Efficiency Score**, takes into account both the valid execution accuracy and the relative efficiency of the predicted SQL compared to the ground truth SQL. It is essential to note that the ground truth SQL represents a query that produces correct results but may **NOT** necessarily be the most efficient SQL. To calculate VES, we first multiply the execution accuracy (whether the predicted SQL produces the correct result) by the relative efficiency score R. R measures the efficiency of the predicted SQL relative to the ground truth SQL in terms of execution time. Thus, VES benefits models that generate accurate and efficient SQL queries compared to the ground truth SQL.
>
>
> **Q4: Clarification on Annotation Roles and Expert Corrections on LLM-generated SQL queries.**
>
> We appreciate your attention to detail, but it seems there may have been a misunderstanding regarding the annotation process described in Section 3.4. To clarify, the expert corrections discussed in this section are applied to the SQL queries and evidence descriptions written by **human** in the annotation process, not the outputs of large language models (LLMs). The annotators were humans crowdsourced for the task. During the annotation process, experts made corrections such as adjusting conditions in questions so the human-authored SQL would execute correctly, fixing insufficient or incorrect external knowledge evidence provided by the human annotators. Moreover, it is crucial to note that our evaluation process does not involve any human intervention, which is all automated. Thus, there is no concern regarding "penalization" within this context. The terms "experts" and "specialists" share the same meaning and refer to the same group of people. To improve clarity, we will consistently use the term "expert" throughout the presentation.
>
> **Q5: Considering prompt costs when electing the most suitable LLM**
>
> We would like to clarify that we did not use any LLMs to assist annotators in generating data. All text, SQL queries, and external knowledge evidence are entirely produced by humans. Thus, we think there's no need to "select" LLMs. However, we appreciate your suggestion and will consider incorporating prompt costs into our evaluation of method performance.
>
> **What does the single parameter θ mean? The entire collection of parameters required to train the LLM?**
>
> The θ represents all the parameters of the text-to-SQL model.
> For T5-based models, full parameters training are required, and for ICL-based models, such as ChatGPT, the parameters from pre-training are frozen.
>
> **Q6: In section 3.3, the 8 math operations are split over the main body and a footnote, when they could all be listed in the same location.**
>
> Our intention was to distinguish between complex operations and basic aggregations commonly found in other text-to-SQL datasets. We understand that this may have caused confusion, and we will incorporate these distinctions into the main text for improved clarity.

---

> > ### Comment · Reviewer_fFFH · 2023-08-23
> > **Commendable work to address the feedback**
> >
> > I thank the authors for the very detailed and thorough comments and in particular for the clarification of points Q3 and Q4.
> >
> > As all my concerns have been addressed and the same degree of effort was put in the responses to all other reviewers, I will be positively revising my rating.

---

> > > ### Author Response · Authors · 2023-08-29
> > > **Thanks for your feedback**
> > >
> > > We greatly thank your valuable feedback and suggestions for our work and revision of ratings. We are hope that BIRD can pave the way for the successful implementation of LLMs as reliable assistants in database interfaces in the future.

---

### Official Review · Reviewer_3iDu · 2023-07-23
**The BIRD text-to-sql benchmark presents more complex queries, diverse databases, background knowledge and an efficiency metric**

**Rating:** 7
**Confidence:** 4
**Correctness:** Evaluation and code seem correct.

**Strengths:**

- Realistic benchmark for text-to-sql with more complex queries and databases that cover a variety of domains, which is a much needed contribution.
- BIRD introduces background knowledge e.g. schema information that is shown to improve text-to-sql methods. BIRD also introduces an efficiency metric for evaluating the efficiency of generated SQLs, which is an important novel aspect to evaluate methods on.
- The experiments and analysis support the motivation of the benchmark, e.g. efficiency metric and external knowledge, and compare the content of the benchmark and model performances on the most commonly used existing benchmark, Spider.

**Additional Feedback:**

The supplemental material provides very useful detailed insights, it could be helpful to add some of the figures (e.g. Figure 6, showing the external knowledge) into the main body of the paper.

**Clarity:**

- The introduction could be written more clearly, it now includes a few vague statements such as “mysteries hidden behind database values require external knowledge and reasoning to reveal”. It is recommended to be more concrete on why larger databases are needed to evaluate the performance of text-to-sql methods.
- The process of collecting annotated natural language and SQL queries seems solid, but no details are provided on which platform was used to collect these and how crowd workers were compensated.

**Documentation:**

- The code is well written and open-source, and comes along with a README. This documentation explains how models can be evaluated with the two metrics on the dataset, and how the considered benchmark methods can be trained.
- The submission instructions are limited beyond an email address where people can submit their solution to, although it seems limited to openai-based solutions? There is no plan provided for making sure and the process relies on the responsiveness of a given email, which reduces trust in long-term maintenance of this benchmark (adding new methods over the coming years requires effort).

**Ethics:**

- No issues beyond lacking details on crowdsourcing setup: compensation and platform/crowdworkers considered.

**Limitations:**

Comments on limitations, e.g. different query engines, are included in supplemental material, but not in the main paper.

**Opportunities For Improvement:**

- The main limitation is that the current benchmark does not include state-of-the-art specialized text-to-sql methods (e.g. the SOTA performance on Spider -https://yale-lily.github.io/spider- is .85, not .74). Instead the benchmark considers basic though conventional T5 and GPT-based methods. I strongly recommend adding benchmarking results for the state-of-the-art methods, and recalibrate the analysis with respect to these models. This would improve my rating.
- The analysis of the performance difference on Spider versus BIRD indeed suggests the higher difficulty of the included databases and queries, but it does not support the claim that “large database values” (does this refer to DB dimensions?) require external knowledge: the effect of external knowledge does not necessarily address the challenge of larger databases. With that, the analysis does not sufficiently convince that the dimensions/domain-coverage of databases included in BIRD indeed present a new challenge to methods, besides a statement on this in the error analysis which is not supported by evidence.

**Relation To Prior Work:**

- Insufficient integration of state-of-the-art methods in the benchmark.
- Sufficient coverage of related benchmarks, although DuSQL seems missing in Sec 7.
- Nice comparison with existing benchmark Spider in the analysis of query complexity and performance comparisons of implemented benchmark methods.

**Summary And Contributions:**

**Update 21/08/2023**: I increased my rating from 5 to 7 after the authors' rebuttal/revision, in particular, the inclusion of the SOTA text-to-sql model on Spider, clarifications regarding the annotation protocol, and additional analysis.

--

The submission presents a new benchmark, BIRD, for evaluating text-to-sql methods on various larger databases, and consists of a large number of queries and databases covering different domains and dimensionalities. The work also proposes a new metric that focuses on the efficiency of queries generated by the method, which is considered an open-standing research problem. BIRD is shown to provide a complementary benchmark dataset in comparison to e.g. the Spider benchmark. BIRD also integrates different sources of background knowledge to improve methods. The benchmark includes two LLM-based approaches for text-to-SQL on accuracy and efficiency, and derives directions for future work from efficiency and error analyses.

---

> ### Author Response · Authors · 2023-08-20
> **Thank you for your comments and suggestions**
>
> We are grateful for your valuable feedback and suggestions. In response to your only concerns regarding our paper, we made our best effort to address it below. If our explanation proves to be satisfactory and sufficient, we will be grateful if you consider improving your score.
>
>
> **Q1: Involve experiments of state-of-the-art models**
>
> We thank your suggestion. Please refer to **General Response Q2**. We already add experiments for more advanced models.
>
>
> **Q2: Is there convincing evidence proving that large database values require external knowledge?**
>
> With the increasing size of databases, much of the information gets compressed because it's difficult to unfold all the full context in large database values [1]. Therefore, numerous values are simplified into abbreviations, such as `POPLATEK TYDNE`, as mentioned in the Figure 1. Additionally, real-application questions especially about data analysis involve reasoning beyond basic database content. For instance, in `professional_basketball`, columns display a player's weight and height, but some questions involve calculating the `BMI` using these values. Similarly, the second example in Figure 1 highlights that the database provides bank account type information, while annotators need to determine the number of accounts eligible for loans, which is difficult to efficiently incorporate into large structured databases. Therefore, we deem that external knowledge evidence is crucial for both humans and machines to generate accurate SQLs when handling large databases and tackling more realistic questions since it can provide critical context about values for accurate reasoning. In this paper, we realize and conclude four prevalent types of external knowledge evidence and annotate them into each example as described in Section 3.3. Our experiments, as shown in Table 2 & 3, validate the importance of external knowledge for reasoning about large databases. Providing our annotated knowledge evidence, all models including LLMs (GPT-4, ChatGPT) or smaller models (T5) significantly improve accuracy on BIRD, directly showing its importance for interpreting complex values and answering real-world analysis questions.
>
> [1] Alsberg et al., Space and Time Savings through Large Data Base Compression and Dynamic Restructuring.
>
> **Q3: The platform and compensation of Crowdsourcing**
>
> Please refer to **General Response Q4** for more details. Thanks
>
> **Q4: Comments on limitations, e.g. different query engines, are included in the supplemental material, but not in the main paper.**
>
> We will add this to the main content of the paper in the camera-ready version since it has one additional page to accommodate such additional information.
>
> **Q5: Document about Submission Instructions**
>
> There are always members of our team working in shifts to handle email requests, including answering questions from readers interested in our work and processing model submissions. We support various model formats and allocate corresponding resources, such as an A100 80G, AMD CPU for smaller models (e.g., T5-based), and a parallel GPU Cluster for open-source LLMs (e.g., WizardLM, StarCoder, Vicuna). We also offer closed LLM API keys for ICL-based methods.
>
> Due to NLP advancements, it is challenging for users to pre-deploy methods on Codalab since its GPUs are insufficient for text-to-SQL models, including T5-3B-based methods, let alone open LLMs. Therefore, we use email for efficient contact and resource allocation.

---

> > ### Comment · Reviewer_3iDu · 2023-08-21
> > **Thorough implementation of the provided feedback**
> >
> > Thank you for the elaborate response(s). I appreciate how you have addressed my main concerns regarding inclusion of SOTA text-to-sql models (with DIN-SQL), while also providing some additional valuable analyses as suggested by other reviewers. The much lower execution accuracy of DIN-SQL on BIRD compared to Spider illustrates the new challenges introduced by the BIRD benchmark. I am confident of the significant contribution of BIRD and will adjust my rating accordingly.

---

> > > ### Author Response · Authors · 2023-08-29
> > > **Thank you for your reply**
> > >
> > > We are grateful for your insightful suggestions and timely feedback!

---

### Official Review · Reviewer_YmxV · 2023-07-23
**Solid benchmark paper with comprehensive evaluation**

**Rating:** 8
**Confidence:** 5
**Correctness:** Yes.
**Clarity:** Yes.

**Strengths:**

1. It is clear that a lot of annotation effort and quality control have been put into the benchmark construction. Compared to previous large-scale academic text-to-SQL benchmarks, BIRD introduces several challenges that are closely related to real-world scenarios. I believe this is a highly valuable benchmark for the community.

2. The evaluation of SQL efficiency is new in the text-to-SQL domain. This evaluation further demonstrates that existing text-to-SQL models are still far from optimal regarding generating efficient SQL queries.

3. The authors curated the test data by the specialized team, so that the test set is less likely to be exposed to LLM training.

4. There is an extensive evaluation of different models and approaches for text-to-SQL generation. While on Spider, the performance of fine-tuned T5 models is not much worse than prompting with GPT models, it is interesting to see that GPT models perform much better on BIRD than finetuned models.

**Additional Feedback:**

See the opportunities for improvement section for questions.

**Documentation:**

Yes.

**Ethics:**

No.

**Limitations:**

The authors have adequately addressed the limitations and potential negative societal impact of their work.

**Opportunities For Improvement:**

I have a few questions about the data split and baseline evaluation.

1. How are the databases split among training, development and test sets? Specifically, Figure 3 (a) shows the domain of each database, and I wonder whether the development and test databases are drawn from different domains to the training databases, or the domains of development and test databases are also covered in the training set?

2. Have the authors evaluated GPT-4 on BIRD? It is interesting to see if GPT-4 can outperform other models.

3. I think few-shot prompting has the potential to improve the performance of GPT models, especially for Codex. For example, [1] shows that Codex achieves high accuracy with few-shot prompting. It will be interesting to see whether adding more exemplars in the prompt improves the performance.

4. The paper discusses the challenges of understanding noisy data values. From my understanding, the prompt used in this work does not include database values, and instead only uses the database schemas. Both [1] and [2] show improved results when 1 or multiple data rows are added to the prompt. It would be nice to see the results of adding some data rows in the prompt.

[1] Chen et al., Teaching Large Language Models to Self-Debug.
[2] Rajkumar et al., Evaluating the Text-to-SQL Capabilities of Large Language Models.

**Relation To Prior Work:**

Yes.

**Summary And Contributions:**

This work presents BIRD, a new benchmark for text-to-SQL generation. Compared to previous large-scale academic text-to-SQL benchmarks such as Spider and WikiSQL, BIRD introduces more realistic and noisy database values, which then requires external knowledge grounding to provide more background to understand the database schemas and values. This aspect is related to some recent text-to-SQL benchmarks such as KaggleDBQA, but BIRD is of a much larger scale. Meanwhile, BIRD evaluates the generated SQL efficiency, and to my knowledge, this work conducts the first evaluation on this metric. The evaluation includes the state-of-the-art text-to-SQL models on Spider that perform supervised training with T5 models, and in-context learning with code-davinci-002 and gpt-3.5-turbo. The results demonstrate that BIRD is a challenging benchmark for all existing text-to-SQL models, and there is a huge gap to human performance.

---

> ### Author Response · Authors · 2023-08-20
> **Thank you for your appreciation and comments**
>
> Thanks for your valuable suggestions and comments.
>
> **Q1: How are the databases split among training, development, and test sets? Specifically, whether the development and test databases are drawn from different domains to the training databases or the domains of development and test databases are also covered in the training set?**
>
> Thanks for your question. BIRD aims to evaluate the generalization of models to unseen databases. To ensure an accurate evaluation of semantic parsing capabilities when dealing with complex and large database values instead of memorizing some specific domain knowledge, we purposefully avoid including the same domains from the training set in the development or testing sets.
>
> **Have the authors evaluated GPT-4 on BIRD? It is interesting to see if GPT-4 can outperform other models.**
>
> Thanks for pointing out this. The GPT-4 keys were not available when we did the experiments. But now we update results for most recent and very powerful LLMs such as GPT-4 (`gpt-4-32k`), Claude-2 (`claude-2`), Palm-2 (`text-bison-001`) as shown in **General Response Q2**. Additionally, we also implement a SOTA model from spider leaderboard to our dataset based on GPT-4. We have updated learderboard [https://bird-bench.github.io/](https://bird-bench.github.io/) and would update these results in the revision of our paper.
>
> **Q2: Few-shot prompting has the potential to improve the performance. It will be interesting to see whether adding more exemplars in the prompt improves the performance. Also prompt containing few rows of value can help model fix problems of dirty data types**
>
> Yes, it's interesting. In our recent experiments and leaderboard, we implement a state-of-the-art model in the Spider, DIN-SQL[1], decorating such prompting skills on top of GPT-4. The prompt of this model employs three rows for each database, encompassing few-shot examples of schema linking, difficulty classification, and self-corrections, etc. The experimental results reveal that these approaches are effective in the BIRD dataset; however, they still fall short of human performance. This suggests that further investigation is needed, potentially focusing on database retrieval or more advanced reasoning methods, in order to address scenarios involving extensive and complex database values.
>
> [1] Pourreza et al. DIN-SQL: Decomposed In-Context Learning of Text-to-SQL with Self-Correction. 2023.

---

> > ### Comment · Reviewer_YmxV · 2023-08-30
> >
> > Thanks for the response and adding baseline results. I keep my score.

---

> > > ### Author Response · Authors · 2023-08-30
> > > **Thanks for your reply**
> > >
> > > Thank you for your awesome suggestions and appreciation of our work!

---

### Official Review · Reviewer_QBSP · 2023-07-26
**A beneficial dataset and benchmark for text-to-SQL parsing resembling closer to real-world data and incorporating efficiency metric**

**Rating:** 8
**Confidence:** 3
**Correctness:** Yes.

**Strengths:**

* The proposed benchmark scheme takes execution efficiency into consideration which is an important addition to the current evaluation scheme.
* Curated dataset is featured as large-scale, multi-domain coverage and various question/query categories, resembling closer to real-world scenario compared to currently available alternatives.


**Additional Feedback:**

The authors address an interesting and challenging problem with this dataset and benchmark.

**Clarity:**

Overall, the paper is well written, but occasionally a bit colloquial. Structure and reasoning is rather clear, with minor room for improvement (see “Limitations”).

**Documentation:**

The documentation appears extensive and comprehensive. I would like to see a section on how to contribute to this with additional use cases, e.g. a minimal set of information required to extend the database schemas/tables/values.

**Ethics:**

I assume the data is artificial and thus does not inadvertedly expose real names etc.

**Limitations:**

* The questions are classified into fundamental type and reasoning type, would be nice to see benchmarking results detailed on each category, preferably both macro- and micro-categories.
* In section 6.3, it is not clear what is the purpose of mentioning Two-stage optimization, and how it is connected to BIRD. In addition, more explanation and results should be presented to elaborate the significance and efficacy of “Chat w/ Database”
* Some  potential improvements can be done for reader-friendliness, to name a few: a) Figures for illustrative purposes, e.g. Figure 1 and 2 are difficult to read due to small text size. b) Are results in table 2 and 3 percentage? Would be nice to note down. c) Line 195, typo tex-to-SQL. d) Line 194-196 seems like the two scenarios are mixed up in terms of which one requires self-knowledge grounding. e) Line 208 mentioned a non-existing table 4.

**Opportunities For Improvement:**

* Although each database is significantly bigger compared to other benchmark datasets, the total number of databases is somewhat limited
* Quite some domains are only presented with rather small size databases


**Relation To Prior Work:**

Prior work is clearly introduced and compared against.

**Summary And Contributions:**

This paper presents a new dataset for benchmarking Text-to-SQL parsing performance in terms of both correctness and efficiency. The dataset features in large-scale, various question/query categories, wide domain coverage and supplemented with external knowledge. In answering the question “Can LLM serve as a database interface”, baseline models created by fine-tuning with TE and in-context learning with LLMs are tested against the dataset evaluated by Executive accuracy and valid efficiency score, revealing that even the state-of-art LLMs are still facing large performance gap to humans in Text-to-SQL parsing task.

---

> ### Author Response · Authors · 2023-08-20
> **Thanks for your time and comments**
>
> We are glad that you like our work and very grateful about your insightful comments. We address your only concerns about this paper by following Q-A pairs:
>
> **Q1: Although each database is significantly bigger compared to other benchmark datasets, the total number of databases is somewhat limited**
>
> We appreciate your concerns regarding the limited number of databases in our benchmark. Actually, it's extremely hard to collect databases with large and complex values under the appropriate open-source licenses. We also selected databases carefully to represent a wide range of domain areas. We finally kept 95 databases and it stands as the largest dataset currently available in text-to-SQL domain.
>
> **Q2: Quite some domains are only presented with rather small size databases**
>
> There is an imbalance in database sizes among the BIRD domains due to the inherent challenges in collecting real-world databases across domains. Our main goal is to provide a comprehensive collection of real and complex databases, more than alignment of sizes across domains since we concern this will break the real distributions. Despite their size, these smaller databases still require domain expertise and deductive reasoning, which are essential for the development of text-to-SQL systems. Therefore, we also keep these databases in BIRD. However, we agree that aligning the database sizes across domains is a promising avenue for future advance. Thanks for your suggestions.
>
> **Q3: The questions are classified into fundamental type and reasoning type, would be nice to see benchmarking results detailed on each category, preferably both macro- and micro-categories**
>
> Thanks for your suggestion, we have conducted this experiment for each category of question types (in **General Response Q3**) and will add this to revision of papers. And we will even provide this to challenge takers who will submit their methods or models to BIRD-test for their better understanding of pros & cons of their models.
>
> **Q4: In section 6.3, what is the purpose of mentioning Two-stage optimization, and how it is connected to BIRD. In addition, more explanation and results should be presented to elaborate the significance and efficacy of “Chat w/ Database”**
>
> In our discussion in section 6.3, we demonstrated two-stage optimization and "Chat with Database" as methods to improve text-to-SQL efficiency, which are crucial factors for BIRD's large real-world databases. SQL efficiency is crucial considering BIRD's emphasis on large datasets. These are two promising directions for future research in the field of text-to-efficient SQLs that we propose.
>
> Two-stage optimization is a straightforward but effective method for producing SQL queries with high performance. For example, a two-stage optimization approach for improving SQL efficiency might encompass two stages: SQL generation and query execution optimization. In the examples examined, two-stage optimization reduced execution time by 77.5% while maintaining accuracy.
>
> "Chat with Database" enables models to interact with the database, allowing them to better comprehend characteristics such as data types, value distributions and optimize SQL execution efficiency as a result. For example, ChatGPT can improve execution efficiency by 87.3%, by adding indexes to databases after knowing its value distribution through interaction.
>
> **Q5: Some potential improvements can be done for reader-friendliness**
>
> Thanks for your suggestions, we will modify these flaws in revision.

---

> > ### Comment · Reviewer_QBSP · 2023-08-29
> > **Nice work**
> >
> > I thank the authors for their effort and time on this very relevant dataset and bechmark. I stand with my rating and am confident that this an important contribution to the field. In case the authors have missed it, this may be a good next candidate the extend the benchmark: https://github.com/defog-ai/sqlcoder

---

> > > ### Author Response · Authors · 2023-08-30
> > > **Thanks for your appreciation and suggestions**
> > >
> > > We greatly appreciate your feedback and timely suggestions. We also notice the recent release of CodeLLMs, which have the potential to be evaluated on BIRD, such as SQLCoder (Aug. 21st), Lemur (Aug. 23rd), and Code Llama (Aug. 26th). Our research is an ongoing, long-term endeavour, and we intend to evaluate these advanced LLMs in the near future. Thank you again.

---

### Official Review · Reviewer_Kfee · 2023-08-02

**Rating:** 8
**Confidence:** 4

**Strengths:**

- With the emergence of Large Language Models (LLMs), the text-to-SQL community has recognized a need for more challenging benchmarks. Although several recent works have acknowledged this need and proposed challenging benchmark datasets, no general-purpose, large-scale cross-domain benchmark is currently available for the community. To fulfill this need, the authors have proposed a new dataset, aiming to reflect the demands of real-world, large-scale applications.
- The authors have shifted the focus of text-to-SQL tasks to text-to-efficient-SQL, introducing an evaluation metric known as the Valid Efficiency Score (VES). This adds an important dimension of efficiency to text-to-SQL evaluation, essential for practical and large-scale applications.
- Although some details may be missing, the research has been conducted with a rigorous annotation process and detailed experimental analysis. This indicates a high level of quality and careful effort.
- This work includes an interesting approach of comparing model performance with human performance (during the 1st round of the annotation process), providing a valuable benchmark and context for evaluating progress in the field.

**Additional Feedback:**

Here are some line-specific feedback and suggestions:
- (L30): consider replacing "SOTA" with the full term before introducing the acronym to aid understanding.
- (L33): "previous benchmark"; typo
- (L55): consider changing "current models" to "current text-to-SQL parsers" to enhance clarity.
- (Figure 2): consider the consistency between the diagram and caption language (e.g., use either "experts" or "specialists", "Training & Test" or "teach & evaluate", "pass" or "enroll").
- (L61, L65): the term “text-to-SQL” may be ambiguous; consider using the more specific phrase “text-to-SQL task.”
- (L81): consider stating "easily accessible with the appropriate licenses” with Appendix B.9
- (L115): "The more semantic-equivalent and efficient SQL selected"; it seems to me that these two aspects might generally conflict with each other. could you explain which aspect was prioritized in the selection?
- (L148): "deeper" might be more appropriately replaced with "darker."
- (L150): "considerable proportion" needs elaboration; it would be good to explicitly state the proportion (in parentheses). It doesn't appear considerable in terms of the entire data value, but what is the percentage of date-related values in the table compared to the whole table?
- (Figure 4): please ensure order consistency between the diagram and main text, particularly regarding intricacy and diverse patterns.
- (Table 2/3): Is the expression “SOTA text-to-SQL models” appropriate in the caption?
- (L195): “tex-to-SQL”; typo
- (L198-L199): change "the performances of FT models (T5) perform"; typo
- (L233): the term “text-to-efficient-SQL” appears suddenly; please consider adding context by using the phrase “text-to-efficient-SQL conversion” or “text-to-efficient-SQL task.”

**Clarity:**

The overall structure of the paper is generally well-written and easy to follow.

Here are some line-specific concerns:

- (L45-46): do all 80 open-source relational databases come from real analysis platforms like Kaggle and Relation.vit? Does this include databases crafted in the DuSQL style, or are they excluded?
- (Figure 2) should the standard of question annotation entrance is “more than 8” means 9/10 not 8/10? If not, the description in Appendix should be modified.
- (L69-L70): although details about annotation process can be found in the Appendix section, it would enhance understanding to include representative numbers in the main text, such as the number of SQL/NLQ annotation applicants, the number of SQL/NLQ annotators, and the number of experts.
- (L73) “choose to self-design database schemas and value production.” could be more specified.
- (L89-L91): it would be highly helpful to include specific examples related to “full schema names” and “value description.”
- (L91): please clarify the “value description” with more detail; is there a description for each value existing in every column, or are there only descriptions for specific columns, such as a column with a categorical variable?

**Correctness:**

The claims appear valid, supported by evidence. The dataset is constructed soundly, and the evaluation methods and experiment design are generally appropriate and correctly performed. Some minor improvements could further strengthen the work.

**Documentation:**

All documentation-related aspects are sufficient.

**Ethics:**

There are no significant ethical concerns.

**Limitations:**

Even data analysis platforms, which may eventually or potentially have complex schemas and a large scale of values compared to previous benchmarks such as SPIDER, often undergo post-processing from their original data lakes. This is typically done to align with specific goals, such as dataset competition (e.g., Kaggle) or machine learning research.

**Opportunities For Improvement:**

Although the dataset construction process is quite rigorous, there are some missing parts. More details on the following aspects would help to understand the dataset construction process more fully:

- Questions about the database collection:
    - While DuSQL is described as differing from real-world scenarios in the main text, the paper states that 20% of the databases were created using an approach similar to DuSQL. Are there any distinguishing features for large database values between DuSQL and the 20% of databases in BIRD?
- Questions about the annotation process:
    - Some detailed numerical parts are missing. (e.g., What is the number of applicants for question annotation entrance?, What is the number of applicants for SQL annotation entrance?, What is the number of SQL annotators?)
    - Who are the three text-to-SQL experts? Did they also participate in the SQL annotation entrance evaluation or the SQL annotation process?
    - Does the text-to-SQL evaluation process include an assessment of the efficiency of SQL query annotation? If so, how are they evaluated, and is this also done by the three experts?
    - Regarding “database students,” do you mean students studying databases or students in data-related fields? Consider specifying this, such as "students studying in data-related fields."
    - Where are the details about External Knowledge Evidence Annotations? How were they collected, and by whom? Was this done by the Question Annotator or the Experts?

Also, I have several concerns about the Valid Efficiency Score (VES):

- Regarding the function E, the possible range of its results might significantly affect the R value and, consequently, the VES measurement. Is it still stable to use in this context? What is the possible range of E in the current experiments?
- Should the possible range of the R function's results be considered as (0, +inf)? If this range is too broad, it might make comparison difficult using this metric. I'm curious about the range of the R value in the current experiments.
- What is the specific method for removing outliers in VES?
- When measuring VES, what is the standard deviation (STD) if attempted 100 times (or after removing outliers)?
- The function E is defined as absolute execution efficiency, but is a higher value more efficient, or is a lower value more efficient? Or is the definition situation-dependent? If so, does this also apply to the meaning (or formulation) of function R or the VES metric?
- The VES metric includes a ratio that compares the efficiency of each ground truth (GT) SQL query to the corresponding correctly predicted SQL query. The source and nature of these ground truth queries need to be discussed. If the GT SQL query are not optimized, the ratio might favor predicted queries (that are even inefficient) but still better than (a poorly optimized) GT SQL query.
- The use of the square root function is mentioned to "minimize random instances which are abnormally faster or slower than the ground-truth SQLs." This approach aims to reduce the impact of extreme efficiency values, but it also compresses the range of efficiency scores. This makes it harder to distinguish between moderately efficient and highly efficient queries. The rationale for choosing the square root function and its effect on the evaluation results should be explained in greater detail.

**Relation To Prior Work:**

Yes, it is mostly clear.

- In Table 1, the authors provide an overview comparison with other cross-domain text-to-SQL benchmarks.
- In Figure 4, the authors provide a comparative analysis of SQL queries with other cross-domain text-to-SQL benchmarks.
- In Section 7, the authors give a brief overview of prior works, including single-domain, cross-domain, and more recent practical text-to-SQL datasets, thereby highlighting the novelty of this work.
- In Appendix B.11, the authors include a comprehensive analysis of existing text-to-SQL models and their efficiency with respect to SQL.

There are some concerns:

- In Table 1, the term "Function" may require a detailed example or definition (one may ask ”what is the SQL function?”). Clarification of this could make it easier to understand what sets BIRD apart from other datasets.
- In Section 7, a clear differentiation between KNOWSQL and BIRD may be needed, beyond just the language difference (English vs. Chinese). Regarding external knowledge evidence, it might be noted that KNOWSQL has a similar aspect, referred to as a formulaic knowledge bank.

**Summary And Contributions:**

**(25 Aug 2023)**: After reviewing the authors' responses for all reviewers, particularly their comprehensive explanations about the database collection process and clarifications of the evaluation metric (VES), I increased my rating from 7 to 8, as these responses make the work more rigorous.

---

- The paper introduces BIRD, a new benchmark for Text-to-SQL parsing (cross-domain), which comprises 12,751 text-to-SQL pairs, 95 databases with a total size of 33.4 GB, spanning across 37 domains.
- This benchmark presents three distinct challenges aiming to mirror real-world scenarios: handling large-scale databases (which inherently include noisy and dirty values), understanding external knowledge evidence (the mapping from a segment of a NLQ to a SQL segment), and generating efficient SQL queries (a critical aspect when deploying Text-to-SQL model).
- The authors not only focus on the execution accuracy (EX) but also introduce a new efficiency-relative metric for evaluating the running time of generated SQL queries compared to the ground-truth SQL queries.
- The authors highlight that even LLMs perform significantly below human-level performance, achieving an execution accuracy of 40% compared to the human benchmark of 92.86%. Moreover, they provide a comprehensive experimental analysis, including an ablation study of knowledge evidence, efficiency analysis, and error analysis.

---

> ### Author Response · Authors · 2023-08-20
> **Thank you for your detailed reviews**
>
> Thanks for your detailed comments and appreciation!
>
> **Q1: Distinctive features for large database values between DuSQL and the 20% of databases in BIRD? And difference about knowledge bank between KnowSQL and BIRD?**
>
> I would like to clarify that the 20% database for the BIRD benchmark was built using a procedure similar to DuSQL, rather than using the DuSQL database. These procedures are very normal among database creation in DBA. Besides, there are obvious differences in database values between BIRD and DuSQL. To be specific, the differences are:
>
> 1. BIRD contains much larger databases than DuSQL and KnowSQL.
> 2. BIRD includes more diverse and complex data values, such as a higher percentage of date-related values (8% of data). Also, we keep the reasonable size of (**`Null`**).Sometimes (**`Null`**) can reveal external knowledge.
> 3. DuSQL databases are collected from Baidu Baike, which mainly covers the Chinese data and cultures, while BIRD covers the world knowledge.
>
> And there are differences about external knowledge between KnowSQL and BIRD:
>
> 1. Main difference is that KnowSQL only covers the external knowledge over database schemas, such as `winning rate = #won / #games`. However, our benchmark not only considers this but also mainly considers knowledge about values as described in Figure 1, such as `POPLAKE TYDNE refers to weekly issuance` or `Year = 4 refers to the release time is closest`.
> 2. KnowSQL only covers commonsense knowledge and events in China. Our dataset contains world knowledge and data, including different legislations, rules in sports, etc.
> 3. The formulaic bank in KnowSQL is designed to provide clear and concise external knowledge set. In contrast, BIRD employs two types of knowledge representation: knowledge evidence attached to each example (oracle), and knowledge banks embedded within database description CSV files, as illustrated in Figure 2 to simulate real-world scenarios since professional databases typically include a description file detailing schema names, values and external knowledge about this column or value.
>
> **Q3: Who are the three text-to-SQL experts?**
> - A database research scientist who's published over 20 top DB conference papers (e.g., SIGMOD, VLDB).
> - A PhD student with research interests in text-to-SQL, who achieved state-of-the-art results on text-to-SQL open challenges.
> - A DBA engineer with more than 10 years of experience in text-to-SQL applications for both B2B and B2C businesses.
>
> **Q4: Details about Valid Efficiency Score (VES)**
>
> Regarding $E$, in our work, we consider time as the main metric to represent efficiency, where $E ∈ (ϵ, 30s)$. Here, $\epsilon$ is a small positive constant to prevent floating-point overflow. The single $E$ is not stable due to machine status as we discuss in section 5. Lower $E$ refers to faster execution speed, which is more efficient.
>
> Concerning $R$, it represents a normalized efficiency ratio between human-annotated SQL queries and predicted SQL queries to reduce the influence of machine status. The stability of this metric is ensured by running this computation 100 times for each example, filtering outliers, and subsequently computing the average. The outlier refers to data points outside `mean - 3 * standard_deviation` and the upper threshold as `mean + 3 * standard_deviation`. Considering the rapid advancement of technology, it is impractical to estimate the fastest SQL performance. Therefore, the range of $R$, is defined as $R \in (0, +\infty)$. If ${E(\hat{Y}_n)}$ is lower than ${E(\hat{Y})}$, then the relative efficiency score $R$ will be increased. In a short, more efficient.
>
> Even square root function compresses the range of efficiency scores, as shown in the Table 3, the performance difference between various models in terms of VES is obvious. Moreover, the STD of VES on dev set and test set after 10 trials are 0.043 and 0.025 respectively. This proves that VES can stably and obviously measure the overall efficiency difference between Language Model Models (LLMs).
>
> **Q5: In Table 1, the term "Function" may require a detailed example or definition.**
>
> We will include specific categories with examples. The categories will cover:
> - Aggregate Functions, i.e., `MAX()`
> - Window Functions, i.e., `OVER()`
> - Date Functions, i.e., `JULIANDAY()`
> - Conversion Functions, i.e., `CAST()`
> - Math Functions, i.e., `ROUND()`
> - String Functions, i.e., `SUBSTR()`
>
> **Q6: The Roles of GT SQLs in Evaluation of VES**
>
> Indeed, it is important to clarify that the ground truth (GT) SQL in our study refers to human-annotated SQL queries that have been carefully reviewed and selected. This ensures that the calculation of execution (EX) accuracy for predicted SQL queries is reliable, and also a high-quality reference for evaluating the relative efficiency of predicted SQL queries. Without this reference, execution time may vary significantly across different machines or even the same machine with different status.

---

> > ### Comment · Reviewer_Kfee · 2023-08-25
> > **Advancing semantic parsing beyond the SPIDER era**
> >
> > I appreciate the time and effort you've invested in addressing my concerns, as well as those of other reviewers. With further clarification and even slight modifications in the revision, I believe this work will gain increased rigor and credibility. As Reviewer fFFH mentioned, this benchmark dataset has generated significant excitement across various fields. Beyond the SPIDER era, I think this benchmark dataset has the potential to guide the current semantic parsing task toward more practical applications.

---

> > > ### Author Response · Authors · 2023-08-29
> > > **Thanks for your insightful comments and reply**
> > >
> > > We sincerely express our gratitude for your appreciation of our efforts and the time you have dedicated to offering valuable comments & insights to us.

---

### Author Response · Authors · 2023-08-20
**General Response to all Reviewers**

We sincerely appreciate the time and effort all reviewers have invested in examining our paper, and we are grateful for their insightful feedback. We are pleased that reviewers liked the contributions of BIRD. Additionally, before carefully discussing each reviewer's concerns individually, we first summary the following common questions.

**Q1: Typo Errors:**

Thank reviewers for pointing out all minor typos. We will address these in the revision.

**Q2: More Advanced Results:**

> Updated Leaderboard: [https://bird-bench.github.io/](https://bird-bench.github.io/)

We made efforts to implement more advanced models on our benchmark in the last few days. We first invited the authors of the current SOTA method on Spider, namely DIN-SQL based on GPT-4 (`gpt-4-32k`), to submit their solution. It obtains a score of `85.4` on the Spider test dataset and `55.90` on the BIRD test dataset, proving again the challenge of BIRD. Their submission can promise the least prompt design bias for implementation. We also made lots of coding efforts to ensure effectiveness for each example.

In addition, we also test other powerful LLMs, including Palm-2 (`text-bison-001`) and Claude-2 (`claude-2.0`). According to our analysis, GPT-4 is still the best LLM for this challenging text-to-SQL benchmark, followed closely by Claude-2. DIN-SQL prompt involves value sampling (select 3), few-shot demonstration, and self-correction. As presented in the result table, DIN-SQL + GPT-4 outperforms best on BIRD. However, a closer analysis of the results reveals an obvious discrepancy in performance gains between the development (+4.37) and test datasets (+1.01). We hypothesize that it may stem from the overfitting of few-shot demonstration to the development set, thus limiting its effectiveness in generalizing across datasets (test split) [1].

| Models                        | Development Data w/o knowledge | Development Data w/ knowledge | Testing Data w/o knowledge | Testing Data w/ knowledge |
|-------------------------------|:------------------------------:|:-----------------------------:|:--------------------------:|:-------------------------:|
|Palm-2|18.77|27.38|24.71|33.04|  |
|Claude-2|28.29|42.70|34.60|49.02|
|GPT-4|30.90|46.35|34.88|54.89|
|GPT-4 + DIN-SQL| -                              | 50.72| -                          | 55.90                       |
| Human Performance             | -                              | -                             | 72.37                      | 92.96                     |


[1] Levy et al., Diverse Demonstrations Improve In-context Compositional Generalization.


**Q3: Fine-grained Results for micro- & macro Categories:**
Due to space limit, we just show Fine-grained EX results of GPT-4 w/ knowledge here:

| Category           | Simple | Moderate | Challenging | Total  |
|--------------------|--------|----------|-------------|--------|
| Overall           | 54.34  | 34.64    | 31.70       | 46.35  |
|||**Fundamental Type**|||
| Match-based        | 60.64  | 37.37    | 34.52       | 51.44  |
| Ranking             | 32.97| 24.76      | 30.00          | 30.00     |
| Comparison          | 58.44  | 26.09    | 26.67       | 40.34  |
| Counting           | 58.58  | 37.50    | 20.51       | 48.28  |
| Aggregation        | 44.75  | 28.41    | 25.00       | 34.82  |
|||**Reason Type**|||
| Domain knowledge   | 54.60  | 35.17    | 20.41       | 42.02  |
| Numeric computation | 34.78  | 18.89   | 25.00       | 24.47  |
| Synonym            | 53.19  | 43.84    | 25.00       | 46.52  |
| Value illustration | 55.13  | 35.40    | 26.00       | 44.19  |

Based on the given results, we can observe that GPT4 still struggles to perform well in addressing ranking-type and numeric computation questions in text-to-SQL. This observation can encourage further study to enhance these drawbacks.

**Q4: Detailed Information about Annotator Teams & Text-to-SQL experts**

The data is collected from [Alibaba-Appen](https://appen.com/crowd-2/#crowd), an internal version. Each Question annotator receives a `$0.6` reward for each validated question, while SQL annotators earn `$1` per SQL contribution. We also invite Text-to-SQL experts and professors to join to check and annotate external knowledge evidence without compensation. Thanks for their dedication. There are ~1340 SQLs confirmed per week.

During the recruitment phase, we received applications from 154 candidates for question generation and 203 for SQL generation. Ultimately we recruited 11 native speakers for question generation and 20 SQL annotators. Furthermore, question annotators should also provide external knowledge if needed, then experts would select and normalize them into dataset. Database students in this team at least already achieved a good performance in database-related courses.

Text-to-SQL experts just master the recruitment, fixed errors and collected the final data. They don't attend annotation directly.

---

### Decision · Program_Chairs · 2023-09-22

**Decision:**

Accept (Spotlight)

**Comment:**

This work proposes BIRD, a new benchmark dataset for text-to-SQL parsing. Given that the previous benchmarks such as Spider and WikiSQL consist of relatively simple databases (simple schemas, small rows), and the rapidly increasing text-to-SQL performance of recent models such as ChatGPT, the community is in need of a more challenging and realistic dataset. BIRD is certainly a timely contribution, consisting of complex databases from diverse domains, incorporating the need for external knowledge, and the consideration for execution efficiency. This will no doubt generate interest from many researchers in the field of text-to-SQL parsing.